# ORAI2 modulates store-operated calcium entry and T cell-mediated immunity

Martin Vaeth[1], Jun Yang[1], Megumi Yamashita[2], Isabelle Zee[1], Miriam Eckstein[3], Camille Knosp[1], Ulrike Kaufmann[1], Peter Karoly Jani[4], Rodrigo S. Lacruz[3], Veit Flockerzi[5], Imre Kacskovics[4], Murali Prakriya[2] & Stefan Feske[1]

Store-operated $Ca^{2+}$ entry (SOCE) through $Ca^{2+}$ release-activated $Ca^{2+}$ (CRAC) channels is critical for lymphocyte function and immune responses. CRAC channels are hexamers of ORAI proteins that form the channel pore, but the contributions of individual ORAI homologues to CRAC channel function are not well understood. Here we show that deletion of *Orai1* reduces, whereas deletion of *Orai2* increases, SOCE in mouse T cells. These distinct effects are due to the ability of ORAI2 to form heteromeric channels with ORAI1 and to attenuate CRAC channel function. The combined deletion of *Orai1* and *Orai2* abolishes SOCE and strongly impairs T cell function. *In vivo*, *Orai1/Orai2* double-deficient mice have impaired T cell-dependent antiviral immune responses, and are protected from T cell-mediated auto-immunity and alloimmunity in models of colitis and graft-versus-host disease. Our study demonstrates that ORAI1 and ORAI2 form heteromeric CRAC channels, in which ORAI2 fine-tunes the magnitude of SOCE to modulate immune responses.

[1] Experimental Pathology Program, Department of Pathology, New York University School of Medicine, 550 First Avenue, Smilow 316, New York, New York 10016, USA. [2] Department of Pharmacology, Northwestern University, Feinberg School of Medicine, Chicago, Illinois 60611, USA. [3] NYU College of Dentistry, New York University, New York, New York 10010, USA. [4] ImmunoGenes, Makkosi út 86, Budakeszi 2092, Hungary. [5] Experimental and Clinical Pharmacology and Toxicology, School of Medicine, Saarland University, Homburg 66421, Germany. Correspondence and requests for materials should be addressed to S.F. (email: feskes01@nyumc.org).

Ca$^{2+}$ release-activated Ca$^{2+}$ (CRAC) channels mediate Ca$^{2+}$ influx in many cell types. They are formed by the tetraspanning plasma membrane proteins ORAI1, ORAI2 and ORAI3. These ORAI proteins mediate Ca$^{2+}$ influx by store-operated Ca$^{2+}$ entry (SOCE), so named because it depends on the filling state of intracellular Ca$^{2+}$ stores[1]. Upon cell stimulation through receptors that are linked to phospholipase C and the production of IP$_3$, Ca$^{2+}$ is released from the endoplasmic reticulum (ER) via the opening of IP$_3$ receptors. The reduction in the ER Ca$^{2+}$ concentration is followed by activation of two transmembrane proteins located in the ER membrane, stromal interaction molecule 1 (STIM1) and STIM2 (ref. 1). Dissociation of Ca$^{2+}$ from the EF hand domains of STIM1 and STIM2 results in conformational changes that enable them to bind and open CRAC channels in the plasma membrane[1].

CRAC channels are hexameric complexes composed of individual or potentially multiple ORAI homologues[2]. Structure–function analyses of *Drosophila* Orai channels as well as human and mouse ORAI1 have shown that they constitute the pore of the CRAC channel[2–4]. The transmembrane domains are highly conserved between all three ORAI homologues. The first transmembrane domain of ORAI1 lines the channel pore and contains a glutamate residue that constitutes a high-affinity Ca$^{2+}$ binding site and confers strong Ca$^{2+}$ selectivity to the CRAC channel[2,3,5–7]. All three ORAI homologues can function as Ca$^{2+}$ channels when overexpressed[8,9]. The properties of ectopically expressed ORAI1, ORAI2 and ORAI3 channels are similar to those of endogenous CRAC channels[10,11], including activation by Ca$^{2+}$ store depletion, high Ca$^{2+}$ selectivity, inward rectification and Ca$^{2+}$-dependent inactivation (CDI)[1]. However, the three ORAI homologues differ in some of their channel properties, including fast and slow CDI and their sensitivity to pharmacological inhibitors such as 2-aminoethoxydiphenyl borate[8,9].

CRAC currents ($I_{CRAC}$) and SOCE have been reported in many cell types, consistent with the ubiquitous expression of all three ORAI homologues[1,12–14]. Transcript levels of murine *Orai1* are particularly high in immune cells and those of *Orai2* are high in the brain, lung, spleen and small intestine, whereas *Orai3* mRNA is abundant in many solid organs[1,13,15,16]. Expression of *Orai2* is also reported in platelets, melanocytes, B cells, dendritic cells, macrophages and mast cells[13]. ORAI1 is the best-characterized ORAI homologue in terms of its physiological functions, whereas less is known about ORAI2 and ORAI3. Patients with null or loss-of-function mutations in *ORAI1* present with a complex disease syndrome, CRAC channelopathy, which is characterized by immunodeficiency, autoimmunity, muscular hypotonia and ectodermal dysplasia because ORAI1 has critical functions in T cells, skeletal muscle cells, dental enamel-forming cells and eccrine sweat glands[17–19]. ORAI1 is of particular importance for T cell function as emphasized by the lack of CRAC currents and SOCE in T cells of patients with null or loss-of-function mutations in *ORAI1* (refs 17,20,21). The mutations cause a severe combined immunodeficiency-like disease characterized by impaired T cell proliferation, reduced cytokine production, abolished antibody responses and severe viral and bacterial infections[17,18]. In mice, the dependence of CRAC channel function on ORAI1 appears to be less pronounced as naive T cells from *Orai1*$^{-/-}$ mice and *Orai1*$^{R93W}$ knock-in mice (that express a non-functional ORAI1 p.R93W protein) have residual SOCE[22–24], reduced but not abolished cytokine production and the ability (upon differentiation into proinflammatory T helper 1 (T$_H$1) and T$_H$17 cells) to cause experimental autoimmune encephalomyelitis[25,26]. These findings suggest that residual SOCE and T cell function in the absence of ORAI1 may be mediated by ORAI2 and/or ORAI3. Additional functions of

ORAI1 in smooth muscle cells, endothelial cells, platelets, mast cells and secretory cells have been described[19,23,27–30]. ORAI2 and ORAI3, by contrast, are not as well-defined functionally due to the lack of patients with null mutations, gene-deficient mouse models and selective inhibitors of individual ORAI homologues. Whereas ORAI3 has been shown to mediate SOCE in breast, lung and prostate cancer cells and to promote their growth and invasiveness[31], the physiological role of ORAI2 is unclear.

Here we show that ORAI2 is highly expressed in naive T cells, but downregulated in effector T cells, resulting in an increased ORAI1:ORAI2 ratio and stronger ORAI1 dependence in effector T cells. Whereas genetic deletion of *Orai1* reduces SOCE and CRAC currents, deletion of *Orai2* enhances both. These distinct effects are explained by the ability of ORAI2 to form multimeric channel complexes with ORAI1, in which ORAI2 attenuates steady-state CRAC channel currents likely due to its altered inactivation properties compared to ORAI1. Using *Orai1*- or *Orai2*-deficient mice, we find that ORAI1 and ORAI2 mediate SOCE and T cell function. Only combined, but not individual, deletion of *Orai1* and *Orai2* abolishes SOCE completely and interferes with protective antibody responses against viral infection, and prevents autoimmune and alloimmune inflammation in models of inflammatory bowel disease (IBD) and graft-versus-host disease (GvHD), respectively. Collectively, our data demonstrate that ORAI1 and ORAI2 form heteromeric CRAC channel complexes, in which ORAI2 modulates the magnitude of SOCE to control T cell-mediated immune responses.

## Results

**Dynamic regulation of ORAI1 and ORAI2 expression in T cells.** To understand the contributions of the three ORAI homologues to immunity, we analysed the expression of *Orai1*, *Orai2* and *Orai3* genes in various immune cell subsets using public databases (Immgen.org; BioGPS.org: gene 109305 (*Orai1*), 269717 (*Orai2*) and 269999 (*Orai3*)). All three homologues were expressed broadly in immune cells with the highest levels of *Orai1* and *Orai2* found in granulocytes. *Orai2* was expressed above average in T cells compared to most other immune cells (Supplementary Fig. 1a). The analysis of *Orai1*, *Orai2* and *Orai3* mRNA expression in FACS-sorted thymic and splenic CD4$^+$ and CD8$^+$ T cell subsets from wild-type (WT) mice by quantitative real-time PCR (qRT–PCR) confirmed that *Orai1* and *Orai2* are highly expressed in all mature T cell subsets compared to *Orai3* (Supplementary Fig. 1b).

To investigate the role of ORAI2 in T cells, we generated *Orai2*$^{-/-}$ mice by replacing the protein-coding exons of *Orai2* with a LacZ reporter (Supplementary Fig. 2a,b). X-gal (5-bromo-4-chloro-3-indolyl β-D-galactopyranoside) staining of E17.5 heterozygous *Orai2*$^{LacZ/+}$ embryos showed *Orai2* expression in many organs including skin, kidneys and spleen (Supplementary Fig. 2c). FDG (fluorescein di-[β-D-galactopyranoside]) staining of isolated immune cells demonstrated high *Orai2* expression in bone marrow-derived macrophages (BMDMs) (Supplementary Fig. 2d), thymocytes and splenic T cells (Fig. 1a,b). *Orai2* expression was induced in thymocytes of *Orai2*$^{LacZ/+}$ mice from the CD4$^-$CD8$^-$ to CD4$^+$CD8$^+$ stage, consistent with *Orai2* mRNA expression in FACS-sorted thymocytes (Supplementary Fig. 1b). In peripheral T cells, *Orai2* expression was highest in CD44$^{lo}$CD62L$^{hi}$ naive T cells and lower in CD44$^{hi}$CD62L$^{lo}$ effector cells (Fig. 1a,b). Downregulation of *Orai2* was confirmed at the mRNA level (Fig. 1c and Supplementary Fig. 1b). In contrast, *Orai1* mRNA expression was lower in naive than effector CD4$^+$ T cells (Fig. 1c and Supplementary Fig. 1b), resulting in a higher *Orai1*:*Orai2* expression ratio in effector versus naive CD4$^+$ T cells (Fig. 1d). The *Orai1*:*Orai2* mRNA

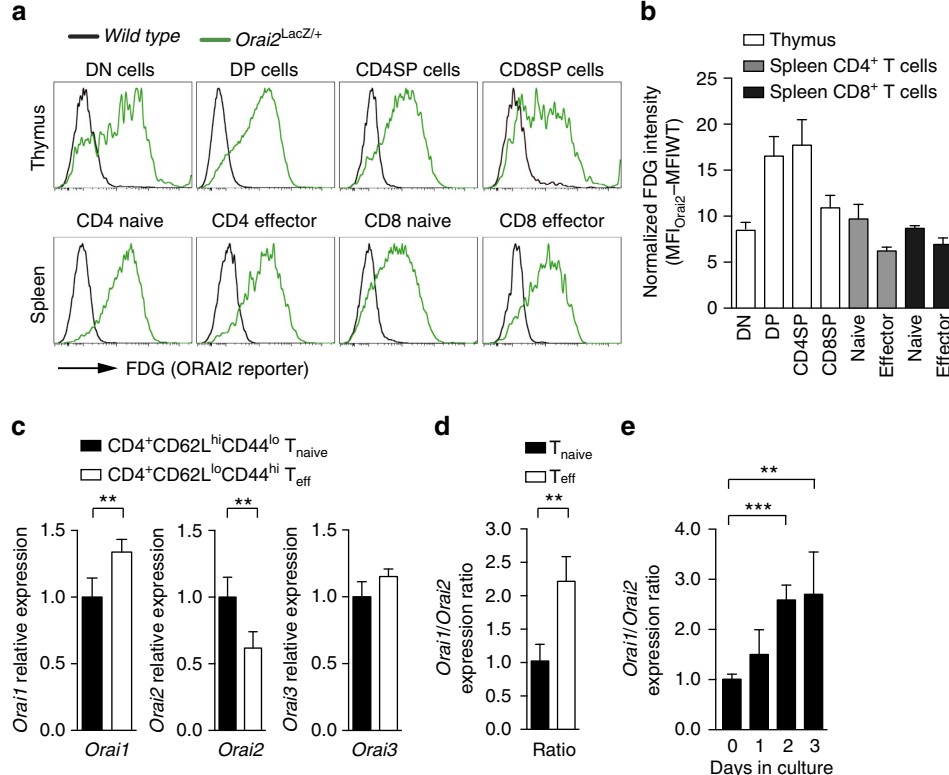

**Figure 1 | Expression of Orai1 and Orai2 genes in T cells.** (**a,b**) LacZ expression (Orai2 reporter) in thymic and peripheral T cell populations of $Orai2^{LacZ/+}$ and WT mice. (**a**) Detection of the green fluorescent LacZ substrate FDG in T cells from WT and $Orai2^{LacZ/+}$ mice by flow cytometry. (**b**) Quantification of normalized FDG fluorescence intensity ($MFI_{Orai2} - MFI_{WT}$) in thymocytes and splenic T cell populations as shown in **a**; means ± s.e.m. of three mice. DN: $CD4^- CD8^-$ double negative; DP: $CD4^+ CD8^+$ double positive; CD4SP: $CD4^+$ single positive; CD8SP: $CD8^+$ single positive. (**c**) Expression of Orai1, Orai2 and Orai3 genes in FACS-sorted $CD4^+ CD62L^{hi} CD44^{lo}$ naive ($T_{naive}$) and $CD4^+ CD62L^{lo} CD44^{hi}$ effector T cells ($T_{eff}$) analysed by qRT–PCR; means ± s.e.m. of four mice. (**d**) Ratio of Orai1 to Orai2 gene expression in FACS-sorted $T_{naive}$ and $T_{eff}$ cells; means ± s.e.m. of four mice. (**e**) Ratio of Orai1 to Orai2 gene expression following anti-CD3/CD28 stimulation of murine naive $CD4^+$ T cells in vitro; means ± s.e.m. of three mice. *$P < 0.05$; **$P < 0.005$; ***$P < 0.001$ in (**c–e**) using unpaired Student's t-test.

expression ratio also increased after activation of naive $CD4^+$ T cells in vitro to generate effector T cells (Fig. 1e). By contrast, the levels of Orai3 mRNA remained unchanged (Fig. 1c). Together, these results show that T cells coexpress ORAI1 and ORAI2, especially in naive T cells, whereas ORAI1 expression dominates in effector T cells, which may contribute to the modulation of SOCE and T cell function.

**ORAI2 controls the magnitude of SOCE in naive T cells.** Homozygous $Orai2^{LacZ/LacZ}$ (hereafter called $Orai2^{-/-}$) mice lacked Orai2 mRNA and protein expression (Supplementary Fig. 2e,f). $Orai2^{-/-}$ mice were born at normal Mendelian ratios without gross morphological or histological abnormalities. To determine the role of ORAI1 and ORAI2 in T cells, we analysed T cells from $Orai1^{fl/fl}Cd4cre$ mice that lack Orai1 in all T cells[25], $Orai2^{-/-}$ mice, and $Orai1^{fl/fl}Orai2^{-/-}Cd4cre$ mice, whose T cells lack both ORAI proteins. When SOCE was induced by depletion of ER $Ca^{2+}$ stores with thapsigargin, which inhibits sarcoplasmic/endoplasmic $Ca^{2+}$ ATPases, ablation of Orai1 reduced SOCE in naive $CD4^+$ and $CD8^+$ T cells by $\sim 30$–50% (Fig. 2a,b and Supplementary Fig. 3a,b) consistent with previous results in T cells from $Orai1^{-/-}$ (refs 22,25,26,32) and $Orai1^{R93W}$ knock-in mice[24]. Unexpectedly, naive $CD4^+$ and $CD8^+$ T cells from $Orai2^{-/-}$ mice had markedly increased SOCE compared to WT cells, whereas combined deletion of both Orai1 and Orai2 completely abolished SOCE (Fig. 2a,b and Supplementary Fig. 3a,b). To determine whether increased $Ca^{2+}$ levels in $Orai2^{-/-}$ T cells were due to increased SOCE,

we analysed $Ca^{2+}$ influx in $Mn^{2+}$ quench experiments. The quench rate of Fura-2 fluorescence in naive $CD4^+$ T cells from $Orai2^{-/-}$ mice was significantly higher than in WT T cells stimulated with thapsigargin in the presence of extracellular $Mn^{2+}$, indicating increased SOCE in $Orai2^{-/-}$ T cells (Fig. 2c). Increased SOCE was observed in naive Orai2-deficient T cells, but not in in vitro differentiated effector T cells. $Orai2^{-/-}$ effector T cells had SOCE comparable to WT cells, whereas deletion of Orai1 resulted in a $>60$% reduction of SOCE (Supplementary Fig. 4a,b) consistent with results in $Orai1^{-/-}$ (refs 22,25,26,32) and $Orai1^{R93W}$ mice[24]. The smaller effect of Orai2 deletion in effector T cells is in line with its reduced expression in effector T cells. These findings suggest that ORAI2 is part of the CRAC channel complex in naive (but not effector) T cells and exerts an inhibitory effect on SOCE.

Increased SOCE was not restricted to $Orai2^{-/-}$ $CD4^+$ and $CD8^+$ T cells (Fig. 2a,b and Supplementary Fig. 3a,b), but was also observed in Orai2-deficient BMDMs (Fig. 3a,b) and BM-derived dendritic cells (BMDCs) (Supplementary Fig. 3c–f). These findings point to a universal role of ORAI2 in attenuating ORAI1-mediated CRAC channel function. This effect is conserved in human fibroblasts because short hairpin RNA (shRNA)-mediated knockdown of ORAI2 caused increased SOCE when compared to fibroblasts transduced with scrambled shRNA (Fig. 2d–f). By contrast, knockdown of ORAI1 in human fibroblasts reduced SOCE consistent with previous observations in fibroblasts from ORAI1-deficient patients[20]. We investigated whether increased SOCE in the absence of ORAI2 is due to

compensatory upregulation of ORAI1. *ORAI1* mRNA levels were comparable to those in control fibroblasts (Fig. 2d). Furthermore, cell surface expression of ORAI1 was unchanged

in shORAI2-treated fibroblasts (Fig. 2g). Likewise, *Orai1*, *Orai3*, *Stim1* or *Stim2* mRNA levels were not altered in primary *Orai2*$^{-/-}$ T cells (Supplementary Fig. 3g). Collectively, these

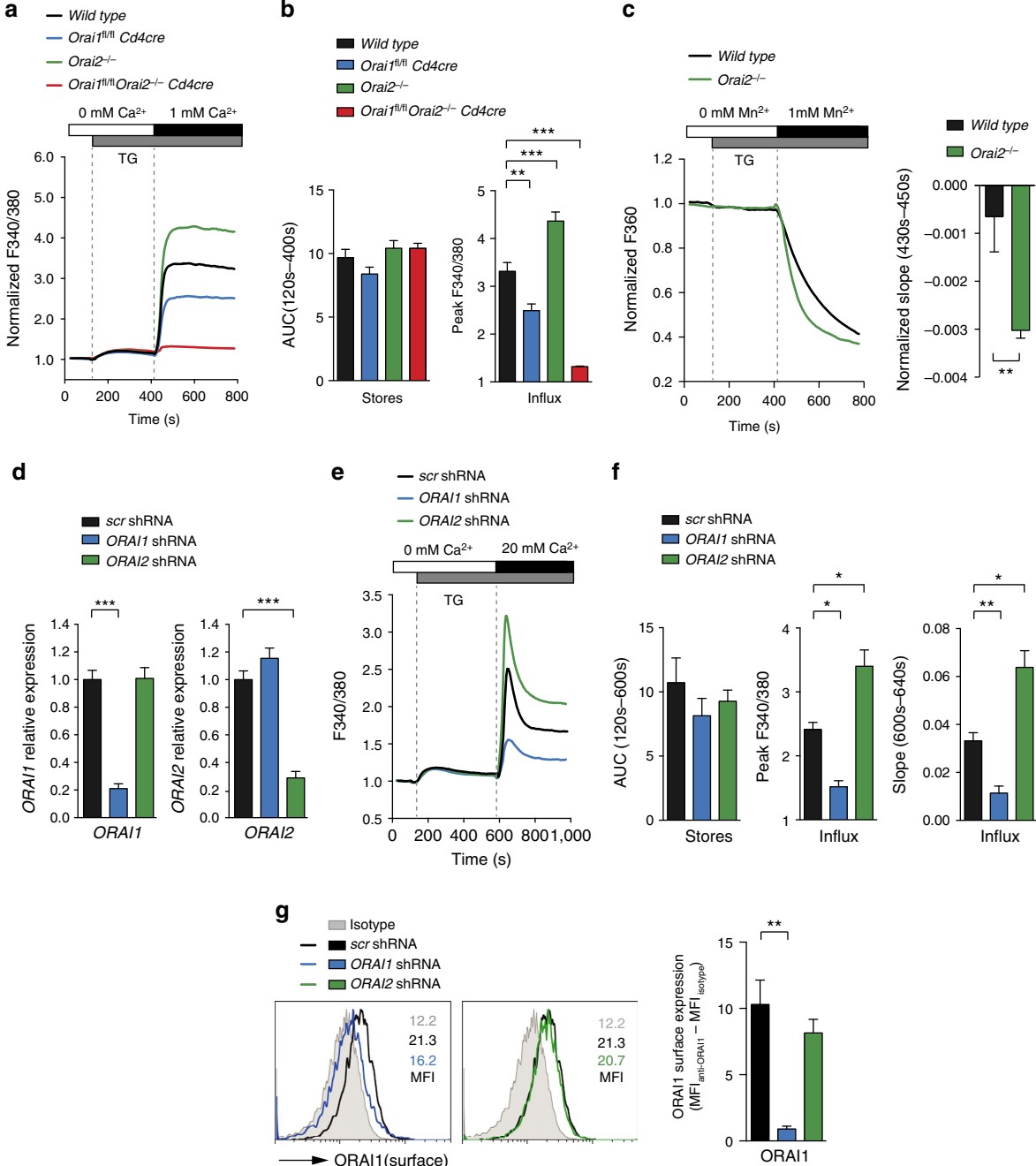

**Figure 2 | Orai2-deficient cells have increased SOCE.** (**a,b**) Analysis of $Ca^{2+}$ store depletion and SOCE following thapsigargin (TG) stimulation in naive T cells from WT, *Orai1*-deficient (*Orai1*$^{fl/fl}$*Cd4cre*), *Orai2*-deficient (*Orai2*$^{-/-}$) and *Orai1*/*Orai2*-deficient (*Orai1*$^{fl/fl}$*Orai2*$^{-/-}$ *Cd4cre*) mice using a FlexStation plate reader. (**b**) Quantification of store depletion (area under the curve, $AUC_{120\,s-400\,s}$) and SOCE (peak F340/380) as shown in **a**; means ± s.e.m. of seven to nine mice. (**c**) Indirect measurement of SOCE by $Mn^{2+}$ quenching of Fura-2 fluorescence. WT and *Orai2*-deficient naive T cells were stimulated with TG in $Ca^{2+}$-free Ringer solution followed by the addition of 1 mM $Mn^{2+}$-containing Ringer solution. Quantification of the slope of $Mn^{2+}$ quenching in TG-stimulated T cells normalized to quenching in non-stimulated cells; means ± s.e.m. of three mice. (**d–g**) Increased SOCE in human fibroblasts after shRNA-mediated knockdown of *ORAI2* is not due to ORAI1 upregulation. (**d**) Analysis of knockdown efficiency of *ORAI1* and *ORAI2* in human fibroblast cells after transduction with shRNAs against *ORAI1* and *ORAI2* by qRT–PCR; means ± s.e.m. of four experiments. (**e**) Analysis of $Ca^{2+}$ store depletion and SOCE following TG stimulation in sh*ORAI1*- or sh*ORAI2*-treated human fibroblast cells using a FlexStation plate reader. (**f**) Quantification of store depletion ($AUC_{120\,s-600\,s}$) and SOCE (F340/380 peak and F340/380 slope$_{600\,s-640\,s}$) in human fibroblast cells as shown in **e**; means ± s.e.m. of four experiments. (**g**) Analysis of ORAI1 surface expression in human fibroblast cells following shRNA-mediated knockdown of *ORAI1* and *ORAI2* as described in (**d**) by flow cytometry using an anti-ORAI1 monoclonal antibody (29A2) recognizing the second extracellular domain of ORAI1; means ± s.e.m. of three experiments. *$P < 0.05$; **$P < 0.005$; ***$P < 0.001$ in (**b–d,f,g**) using unpaired Student's *t*-tests.

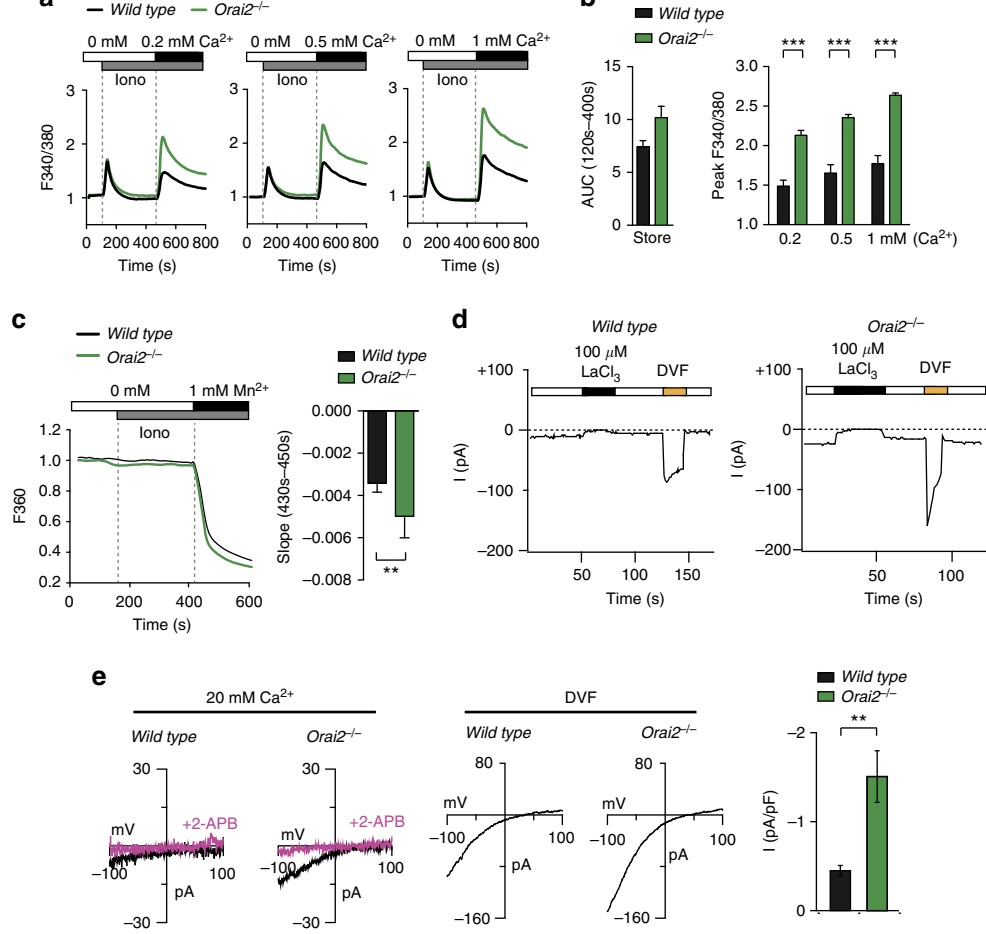

**Figure 3 | *Orai2*-deficient macrophages have increased SOCE and CRAC currents.** (**a,b**) Analysis of SOCE following ionomycin stimulation in WT and *Orai2*-deficient bone marrow-derived macrophages (BMDMs) after readdition of 0.2, 0.5 and 1 mM extracellular $Ca^{2+}$ using a FlexStation plate reader. (**b**) Quantification of store depletion ($AUC_{120\,s-400\,s}$) and SOCE (F340/380 peak) as shown in **a**; means ± s.e.m. of three mice. (**c**) $Mn^{2+}$ quenching of Fura-2 fluorescence to measure store-operated divalent cation influx via CRAC channels. WT and *Orai2*-deficient BMDMs were loaded with Fura-2 and stimulated with 0.3 μM ionomycin in $Ca^{2+}$ and $Mn^{2+}$-free Ringer solution, followed by the addition of 1 mM $Mn^{2+}$. Bar graphs show the quantification of $Mn^{2+}$ quench rates in ionomycin-stimulated BMDMs normalized to unstimulated cells; means ± s.e.m. of three mice. (**d,e**) Electrophysiological properties of CRAC currents in WT and $Orai2^{-/-}$ BMDMs. (**d**) Leak-corrected CRAC currents measured at $-100$ mV are plotted over time. The standard extracellular Ringer solution was periodically switched with a $Na^+$-based DVF solution, revealing permeation of $Na^+$ ions in the absence of extracellular divalents. $I_{CRAC}$ is completely inhibited by 100 μM $LaCl_3$. (**e**) I–V relationship of $I_{CRAC}$ in 20 mM $Ca^{2+}$ (left) and DVF (middle) solutions in WT and $Orai2^{-/-}$ BMDMs and inhibition by 50 μM 2-aminoethoxydiphenyl borate (2-APB). Means ± s.e.m. of the peak current density in WT and $Orai2^{-/-}$ BMDMs; 10–12 cells were analysed (right). \*$P < 0.05$; \*\*$P < 0.005$; \*\*\*$P < 0.001$ in (**b,c,e**) using unpaired Student's $t$-tests.

findings show that increased SOCE in the absence of ORAI2 is not due to compensatory upregulation of ORAI1 or other CRAC channel components but rather indicate a negative regulatory function of ORAI2.

***Orai2* deletion in macrophages enhances CRAC channel function.**
To understand the mechanisms responsible for increased SOCE in the absence of ORAI2, we investigated BMDMs that, like naive T cells, have high ORAI2 expression (Supplementary Fig. 2d–f). Deletion of *Orai2* resulted in elevated SOCE in ionomycin-stimulated BMDMs over a range of extracellular $Ca^{2+}$ concentrations (Fig. 3a,b). It has been suggested that ORAI2 may function as a $Ca^{2+}$ release channel in the ER[33]. *Orai2* deletion might therefore result in increased filling of ER $Ca^{2+}$ stores and potentially affect activation of CRAC channels. Depletion of $Ca^{2+}$ stores with ionomycin in the absence of extracellular $Ca^{2+}$, however, showed only a very moderate increase in $Ca^{2+}$ store content in $Orai2^{-/-}$ compared to WT BMDMs (Fig. 3a,b)

arguing against ORAI2 as an ER release channel. Furthermore, the $Mn^{2+}$ quench rate of Fura-2 fluorescence was significantly higher in $Orai2^{-/-}$ than WT BMDMs supporting the idea that ORAI2 inhibits SOCE through ORAI1 channels (Fig. 3c).

We next analysed CRAC currents ($I_{CRAC}$) in *Orai2*-deficient BMDMs to determine how ORAI2 affects $I_{CRAC}$ (Fig. 3d,e). WT BMDMs had currents consistent with the properties of $I_{CRAC}$ including an inwardly rectifying current–voltage (I–V) relationship, permeation of $Na^+$ seen in a divalent-free (DVF) solution, and blockade by $La^{3+}$. In $Orai2^{-/-}$ BMDMs, these CRAC currents were significantly larger, but otherwise retained the characteristic features of $I_{CRAC}$ in WT BMDMs including inwardly rectifying I–Vs in 20 mM extracellular $Ca^{2+}$ and DVF solutions or $La^{3+}$ blockade (Fig. 3d,e). Enhanced $I_{CRAC}$ in $Orai2^{-/-}$ BMDMs (Fig. 3e) provides direct evidence that the deletion of ORAI2 enhances CRAC channel function in BMDMs. The fact that the properties of $I_{CRAC}$ in $Orai2^{-/-}$ BMDMs were indistinguishable from those of ectopic ORAI1 expression[8,9] suggests that enhanced $I_{CRAC}$ in $Orai2^{-/-}$ BMDMs is

mediated by ORAI1. It is noteworthy that measurements of $I_{CRAC}$ in naive T cells from WT and $Orai2^{-/-}$ mice were not successful due to their small whole-cell currents. Together, we show that ORAI2 attenuates CRAC channel function in BMDMs and naive T cells, presumably by replacing the more efficient ORAI1 subunit in heteromeric ORAI1:ORAI2 complexes.

**ORAI1 and ORAI2 form heteromeric channels**. We next tested if ORAI2 inhibits $I_{CRAC}$ and SOCE by forming heteromeric CRAC channel complexes with ORAI1. We used primary CD4$^+$ T cells from $Orai1^{fl/fl}Cd4cre$ mice, which have reduced SOCE that is mediated by ORAI2 and retrovirally transduced them with either empty vector (EV), WT ORAI1 (ORAI1$^{WT}$) or the non-functional pore-mutant ORAI1$^{E106Q}$ that is expressed in the plasma membrane but unable to conduct Ca$^{2+}$ (ref. 5). We hypothesized that if ORAI1 and ORAI2 form heteromeric complexes, ectopic expression of ORAI1$^{E106Q}$ together with endogenous ORAI2 results in ORAI1$^{E106Q}$:ORAI2 channels that are unable to conduct Ca$^{2+}$. Expression of ORAI1$^{WT}$ (but not EV) strongly enhanced SOCE in $Orai1$-deficient T cells, whereas ORAI1$^{E106Q}$ completely suppressed residual ORAI2-mediated SOCE (Fig. 4a). To confirm this finding, we retrovirally transduced CD4$^+$ T cells from $Orai2^{-/-}$ mice with either EV, ORAI2$^{WT}$ or the non-functional pore-mutant ORAI2$^{E80Q}$ (ref. 14). $Orai2$-deficient T cells transduced with EV had strong SOCE mediated by homomeric ORAI1 channels. Expression of ORAI2$^{WT}$ in $Orai2^{-/-}$ T cells markedly reduced SOCE consistent with the idea that ORAI2 inhibits ORAI1 function, whereas expression of ORAI2$^{E80Q}$ completely abolished SOCE (Fig. 4b). Taken together, these data show that pore-dead mutants of one ORAI isoform suppress the function of the other isoform, likely because ORAI1 and ORAI2 form heteromeric complexes.

An alternative explanation is that overexpression of the ORAI1$^{E106Q}$ mutant in $Orai1$-deficient T cells sequesters the majority of STIM1 required for activation of endogenous ORAI2 (and likewise expression of ORAI2$^{E80Q}$ in $Orai2^{-/-}$ T cells sequesters STIM1 necessary for ORAI1 activation). We therefore mutated leucine (L) 273 in the C terminus of ORAI1$^{WT}$ and ORAI1$^{E106Q}$ to aspartate (D) to abolish ORAI1 binding to STIM1 (refs 34,35). ORAI1$^{L273D}$ was expressed at the plasma membrane (Fig. 4c) but completely abolished SOCE in T cells from $Orai1^{fl/fl}Cd4cre$ mice compared to EV-transduced cells (Fig. 4e,g). SOCE suppression was comparable to that found in $Orai1$-deficient cells transduced with ORAI1$^{E106Q}$ (Fig. 4a,d,g) or the ORAI1$^{E106Q/L273D}$ double mutant that can neither conduct Ca$^{2+}$ nor bind to STIM1 (Fig. 4f,g). These findings argue against STIM1 sequestration by pore mutant ORAI proteins as the cause of suppression of SOCE (Fig. 4a,b), but instead support the model that ORAI1 and ORAI2 form heteromeric channels (Fig. 4h). The fact that ectopic expression ORAI1$^{L273D}$ in T cells of $Orai1^{fl/fl}Cd4cre$ mice almost completely inhibited SOCE is consistent with a report demonstrating that STIM1 binding to all subunits of hexameric ORAI complexes is required for channel activation[36].

To understand if the functional properties of heteromeric ORAI1:ORAI2 channels differ from ORAI1 or ORAI2 homomeric channels, we analysed $I_{CRAC}$ properties in HEK293 cells that coexpressed ORAI1 and/or ORAI2 together with STIM1. Coexpression of STIM1, ORAI1 and ORAI2 resulted in large CRAC currents, which could be blocked with La$^{3+}$ or 2-aminoethoxydiphenyl borate and showed Na$^+$ permeation in DVF solution, thus resembling endogenous $I_{CRAC}$ (Fig. 4i). Furthermore, ORAI1, ORAI2 and ORAI1/ORAI2 channels had similar inwardly rectifying $I–V$ relationships typical of $I_{CRAC}$ (Fig. 4j). Importantly, coexpression of ORAI2$^{WT}$ with the pore-dead ORAI1$^{E106A}$ mutant[5,37] strongly attenuated $I_{CRAC}$ (Fig. 4j,k), confirming our T cell data (Fig. 4a–h). Peak current amplitudes were not different in cells coexpressing STIM1 together with either ORAI1, ORAI2 or both (Fig. 4k), indicating that homomeric and heteromeric ORAI channels have similar abilities to conduct Ca$^{2+}$. By contrast, steady-state $I_{CRAC}$ was significantly reduced in cells coexpressing ORAI1/ORAI2 compared to ORAI1 alone (Fig. 4l). Further analysis of fast CDI of CRAC currents during hyperpolarizing voltage steps showed that ORAI1 currents inactivated by $\sim 20\%$ within 100 ms, whereas ORAI2 currents inactivated more strongly by $\sim 45\%$ (Fig. 4m,n). These findings are consistent with reported differences in fast CDI for ORAI1 and ORAI2 (refs 9,38). Importantly, CRAC currents in cells coexpressing ORAI1 and ORAI2 also had more pronounced fast CDI compared to cells expressing ORAI1 alone (Fig. 4m,n), likely accounting for their smaller steady-state current amplitudes (Fig. 4l). Together, these data show that ORAI1 and ORAI2 form heteromeric channels and suggest that differences in fast CDI between ORAI1 and ORAI2 likely contribute to distinct CRAC current and SOCE levels.

**$Orai1/Orai2$ deletion disrupts peripheral immune homeostasis**. We next investigated the role of ORAI1 and ORAI2 in T cell development and function. $Orai2^{-/-}$ mice had normal populations of immature CD4$^-$CD8$^-$ DN, CD4$^+$CD8$^+$ DP, CD4SP and CD8SP T cells in the thymus and similar T cell numbers in spleen and lymph nodes (LNs) compared to littermates (Fig. 5a,b and Supplementary Fig. 5). Furthermore, the frequencies of CD44$^{hi}$CD62L$^{lo}$ effector CD4$^+$ and CD8$^+$ T cells as well as CD44$^{hi}$CD62L$^{hi}$ memory CD8$^+$ T cells were unchanged in $Orai2^{-/-}$ mice (Fig. 5b). Unperturbed thymic T cell populations, T cell numbers and frequencies of effector and memory T cells were also observed in $Orai1^{fl/fl}Cd4cre$ mice (Fig. 5a and Supplementary Fig. 5a,b).

By contrast, combined deletion of both $Orai1$ and $Orai2$ in T cells caused splenomegaly and lymphadenopathy in $Orai1^{fl/fl}Orai2^{-/-}Cd4cre$ mice at $>3$ months of age with elevated numbers of immune cell populations in the spleen and LNs (Fig. 5a and Supplementary Fig. 5e), which is reminiscent of the phenotype observed in animals with combined T cell-specific deletion of $Stim1$ and $Stim2$ (ref. 39). $Orai1^{fl/fl}Orai2^{-/-}Cd4cre$ mice had no defect in conventional T cell development as thymic cellularity (Fig. 5a) and frequencies of DN, DP, CD4SP and CD8SP T cells were unaltered (Supplementary Fig. 5a,b). In the spleen, the frequencies of T cells were also comparable in $Orai1^{fl/fl}Orai2^{-/-}Cd4cre$ and WT mice. This is consistent with data from $Stim1/Stim2$-deficient mice and ORAI1- or STIM1-deficient human patients, providing further evidence that SOCE is dispensable for conventional T cell development[39,40]. $Orai1^{fl/fl}Orai2^{-/-}Cd4cre$ mice, but not single knockout mice, had elevated numbers of CD4$^+$ and CD8$^+$ T cells with CD44$^{hi}$CD62L$^{hi}$ memory and/or CD44$^{hi}$CD62L$^{lo}$ effector phenotypes, in line with their splenomegaly (Fig. 5a,b). A similar immune phenotype was not observed in either $Orai1^{fl/fl}Cd4cre$ or $Orai2^{-/-}$ mice. The disruption of the peripheral immune homeostasis in $Orai1^{fl/fl}Orai2^{-/-}Cd4cre$ mice was likely due to the decreased numbers of Foxp3$^+$ T regulatory (Treg) cells in the thymus, spleen and LNs compared to WT, $Orai1^{fl/fl}Cd4cre$ and $Orai2^{-/-}$ mice (Fig. 5c and Supplementary Fig. 5e). The development and maintenance of Treg cells is dependent on interleukin-2 (IL-2)[41]. $Orai1/Orai2$-deficient T cells were unable to produce IL-2 $in vitro$ (data not shown), which likely contributed to decreased Treg cell numbers. Collectively, our data show that deletion of $Orai1$ and $Orai2$ together (but not individually) impairs thymic Treg cell development resulting in perturbed immune homeostasis.

**ORAI1 and ORAI2 control immunity to infection**. Patients with loss-of-function mutations in $ORAI1$ that abolish SOCE

exhibit a severe combined immunodeficiency-like disease phenotype with recurrent and chronic viral, bacterial and fungal infections[17,42–45]. ORAI1-deficient human T cells proliferate poorly in response to antigen or mitogenic stimulation and have impaired cytokine production[17,20,21]. The proliferation of

CD4$^+$ T cells from either $Orai1^{fl/fl}Cd4cre$ or $Orai2^{-/-}$ mice after stimulation with anti-CD3/CD28 was comparable to WT cells. By contrast, deletion of both $Orai1$ and $Orai2$ severely compromised proliferation (Fig. 6a and Supplementary Fig. 6). To test if this defect was secondary to decreased IL-2 production

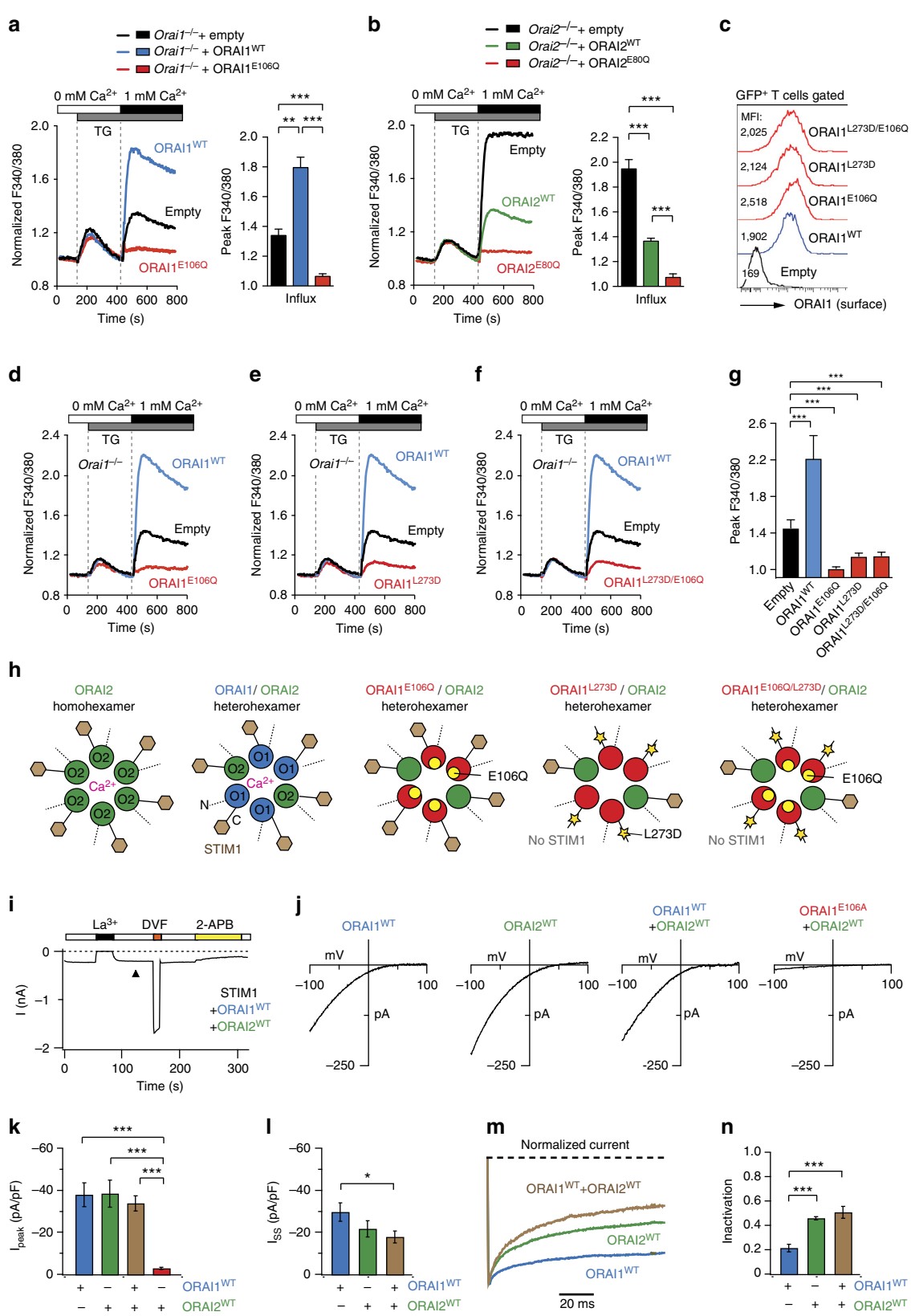

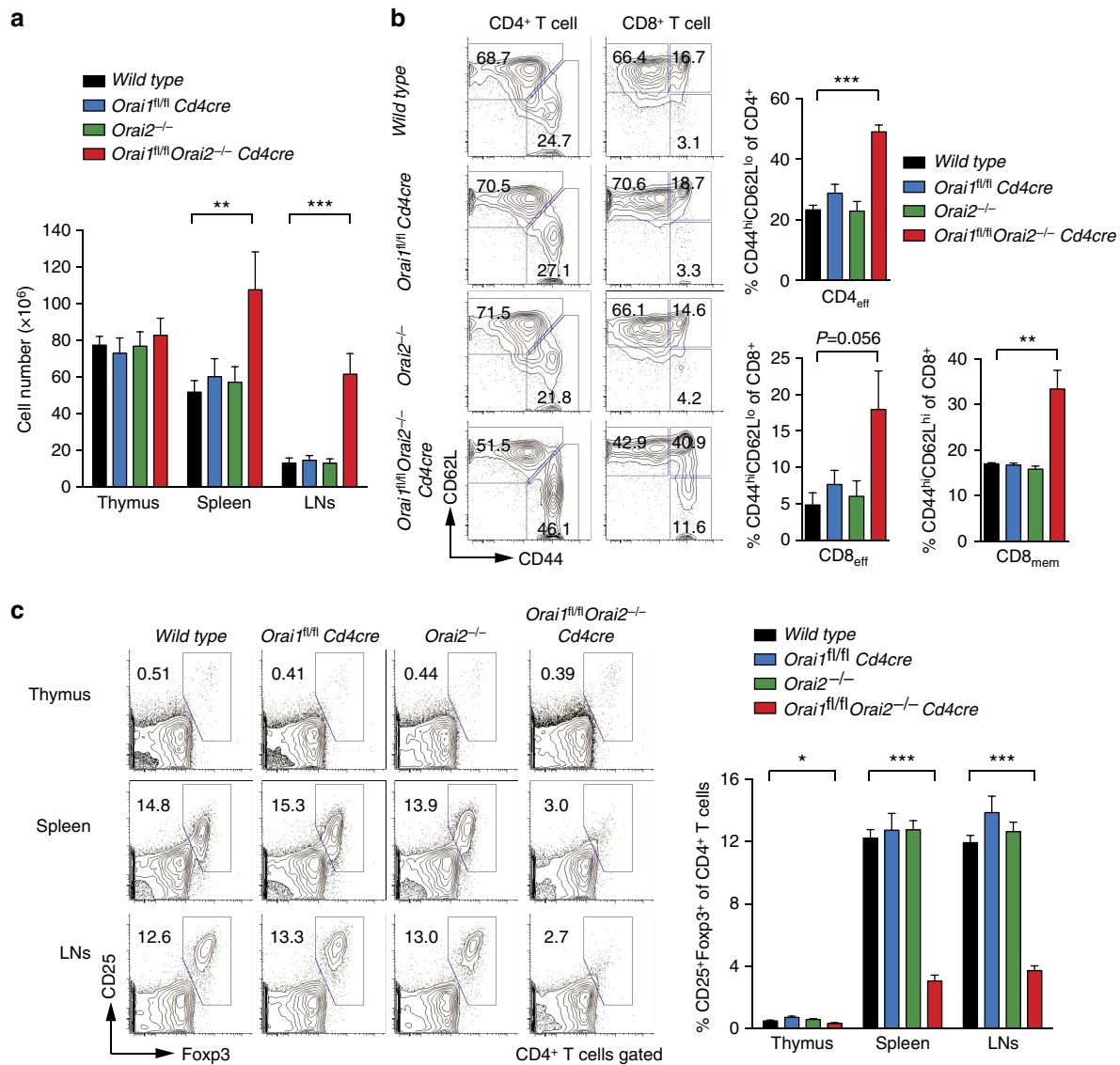

**Figure 5 | Reduced Treg numbers and disrupted immune homeostasis in *Orai1/Orai2*-deficient mice.** (**a**) Cell numbers in the thymus, spleen and LNs of WT, *Orai1*$^{fl/fl}$*Cd4cre*, *Orai2*$^{-/-}$ and *Orai1*$^{fl/fl}$*Orai2*$^{-/-}$ *Cd4cre* (DKO) mice; means ± s.e.m. of 9–11 mice. (**b**) Analysis of CD44$^{hi}$CD62L$^{lo}$ effector CD4$^{+}$ and CD8$^{+}$ T cells and CD44$^{hi}$CD62L$^{hi}$ memory CD8$^{+}$ T cell subsets in the spleen of WT, *Orai1*$^{fl/fl}$*Cd4cre*, *Orai2*$^{-/-}$ and DKO mice by flow cytometry; means ± s.e.m. of four mice. (**c**) Analysis of CD4$^{+}$CD25$^{+}$Foxp3$^{+}$ Treg cells in the thymus, spleen and LNs of WT, *Orai1*$^{fl/fl}$ *Cd4cre*, *Orai2*$^{-/-}$ and DKO mice by flow cytometry; means ± s.e.m. of 9–11 mice. \*$P < 0.05$; \*\*$P < 0.005$; \*\*\*$P < 0.001$ in (**a–c**) using unpaired Student's *t*-tests.

**Figure 4 | ORAI1 and ORAI2 form heteromeric channels.** (**a**) Analysis of SOCE in *Orai1*-deficient T cells transduced with empty vector (EV), wild type ORAI1 (ORAI1$^{WT}$) or a pore-dead ORAI1 mutant (ORAI1$^{E106Q}$). Transduced GFP$^{+}$CD4$^{+}$ T cells were FACS sorted and SOCE was analysed following thapsigargin (TG) stimulation and addition of 1 mM extracellular Ca$^{2+}$. (**b**) Analysis of SOCE in *Orai2*-deficient T cells transduced with EV, ORAI2$^{WT}$ or a pore-dead ORAI2 mutant (ORAI2$^{E80Q}$). GFP$^{+}$CD4$^{+}$ T cells in **a**,**b** were analysed for SOCE (peak F340/380); shown are means ± s.e.m. of cells from three mice measured in triplicates. (**c**) Surface expression of ectopically expressed ORAI1 mutants used in **d–g** using an anti-ORAI1 (2C1.1) antibody by flow cytometry. (**d–g**) Analysis of SOCE in *Orai1*-deficient T cells transduced with (**d**) EV, ORAI1$^{WT}$, ORAI1$^{E106Q}$, (**e**) the STIM1 binding-deficient ORAI1$^{L273D}$ mutant and (**f**) the ORAI1$^{L273D/E106Q}$ double mutant as described in **a**. (**g**) Quantification of SOCE (peak F340/380) as shown in **d–f**; means ± s.e.m. of cells from four mice measured in triplicates. (**h**) Illustration how E106Q and/or L273D mutations affect formation and function of ORAI1:ORAI2 heteromeric channels. (**i**) Time course of CRAC currents in HEK293 cells overexpressing YFP-ORAI1$^{WT}$ and CFP-ORAI2$^{WT}$ together with STIM1 and stimulated with 1 μM TG. Extracellular La$^{3+}$ (150 μM) and 2-APB (50 μM) inhibit the currents in Ringer's solution containing 20 mM Ca$^{2+}$. Application of DVF solution reveals large Na$^{+}$ currents. (**j**) Representative *I–V* relationships of ORAI channels. (**k**) Summary of peak CRAC current amplitudes measured during step pulses to −100 mV in cells expressing the indicated ORAI isoforms. Eight to nine cells per condition were analysed. (**l**) Summary of steady-state (SS) CRAC current amplitudes in cells expressing ORAI isoforms at the end of a 100 ms pulse to −100 mV. (**m**) ORAI1:ORAI2 channels exhibit enhanced fast CDI. Traces depict currents recorded during hyperpolarizing voltage steps to −100 mV. Currents were normalized to their peak values. (**n**) Summary of fast inactivation during the 100 ms pulse measured as: $1 - I_{ss}/I_{peak}$, where $I_{peak}$ and $I_{ss}$ are the peak and steady-state currents during the pulse to −100 mV. **k**,**l**,**n** are means ± s.e.m. \*$P < 0.05$; \*\*$P < 0.005$; \*\*\*$P < 0.001$ in (**a**,**b**,**g**,**k**,**l**,**n**) using unpaired Student's *t*-tests.

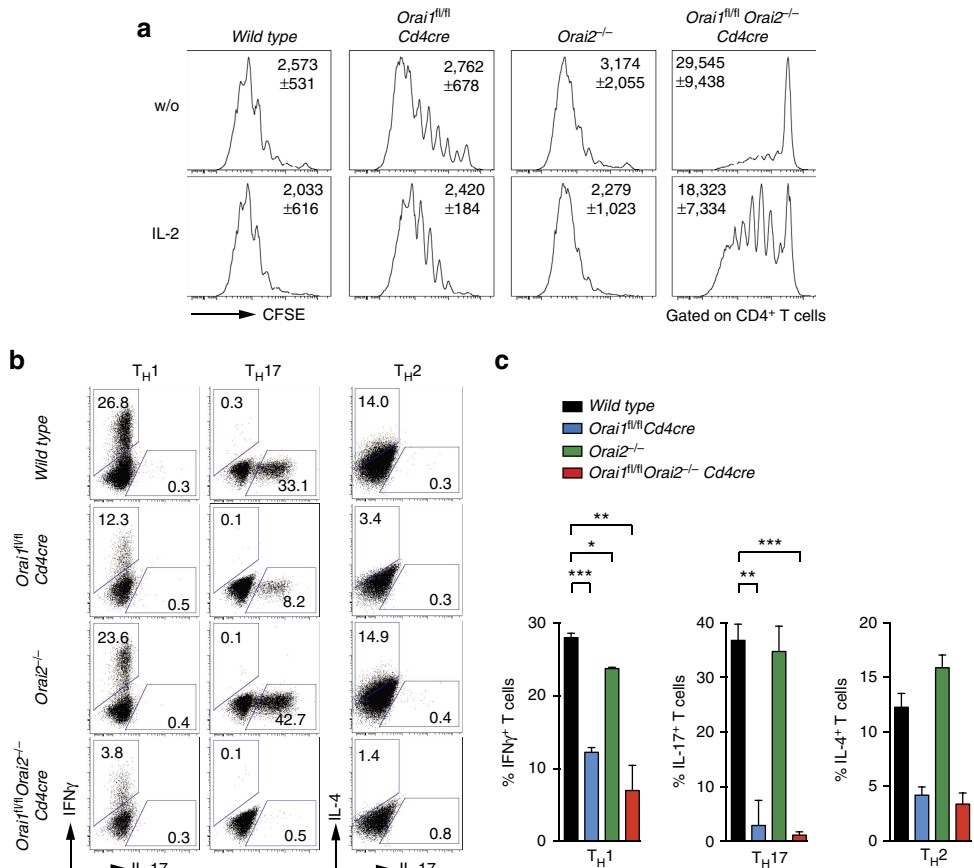

**Figure 6 | Impaired proliferation and cytokine production of *Orai1/Orai2*-deficient T cells *in vitro*.** (a) CD4$^+$ T cells isolated from WT, *Orai1*$^{fl/fl}$*Cd4cre*, *Orai2*$^{-/-}$ and *Orai1*$^{fl/fl}$*Orai2*$^{-/-}$*Cd4cre* (DKO) mice were stimulated with plate-bound anti-CD3/CD28 for 4 days *in vitro* with or without 50 U ml$^{-1}$ recombinant human IL-2 (rhIL-2). Proliferation was analysed by CFSE dilution using flow cytometry; mean fluorescence intensity (MFI) ± s.e.m. of CFSE from two to three experiments is shown within the histograms. Additional time points are presented in Supplementary Fig. 6. (**b,c**) Impaired T$_H$1, T$_H$17 and T$_H$2 cytokine production in *Orai1*- and *Orai1/Orai2*-deficient T cells. (**b**) T cells isolated from WT, *Orai1*$^{fl/fl}$ *Cd4cre*, *Orai2*$^{-/-}$ and DKO mice were differentiated for 3 days under T$_H$1-, T$_H$17- and T$_H$2-polarizing conditions and intracellular IFNγ, IL-17 and/or IL-4 was analysed by flow cytometry after restimulation with phorbol myristate acetate (PMA)/ionomycin for 6 h. (**c**) Quantification of IFNγ, IL-17- and IL-4-positive T cells as shown in **b**; mean ± s.e.m. from three independent experiments (T$_H$1 and T$_H$17) or two experiments (T$_H$2). *$P < 0.05$; **$P < 0.005$; ***$P < 0.001$ in (**c**) using unpaired Student's *t*-tests.

that functions as an autocrine growth factor[41], we stimulated *Orai1/Orai2*-deficient T cells in the presence of recombinant IL-2. However, IL-2 supplementation only partially rescued proliferation in *Orai1/Orai2*-deficient T cells (Fig. 6a), indicating an intrinsic, IL-2-independent role of SOCE in T cell proliferation. Production of cytokines by CD4$^+$ helper T cells (T$_H$) is dependent on SOCE[45] and is impaired in ORAI1-deficient patients[46]. We therefore differentiated naive T cells from WT, *Orai1*$^{fl/fl}$*Cd4cre*, *Orai2*$^{-/-}$ and *Orai1*$^{fl/fl}$*Orai2*$^{-/-}$*Cd4cre* mice into T$_H$1, T$_H$17 and T$_H$2 cells *in vitro* and analysed the expression of interferon-γ (IFNγ), IL-17 and IL-4, respectively (Fig. 6b,c). Cytokine production was significantly reduced in *Orai1*- and *Orai1/Orai2*-deficient T$_H$1, T$_H$17 and T$_H$2 cells, whereas *Orai2*$^{-/-}$ T cells had normal cytokine expression compared to WT cells.

ORAI1 and STIM1-deficient patients have defective humoral immune responses and lack antigen-specific antibodies[17,21]. We recently reported that STIM1 and STIM2 are required for germinal centre (GC) reactions and the production of virus-specific antibodies by controlling the differentiation of T follicular helper (T$_{FH}$) cells[18]. To test how ORAI1 and ORAI2 contribute to T cell-dependent humoral immunity, we infected WT, *Orai1*$^{fl/fl}$*Cd4cre*, *Orai2*$^{-/-}$ and *Orai1*$^{fl/fl}$*Orai2*$^{-/-}$*Cd4cre* mice with lymphocytic choriomeningitis virus (LCMV) and analysed

T$_{FH}$ cells, the GC reaction and anti-LCMV antibody production. At 10 days postinfection, CD4$^+$CXCR5$^{hi}$PD-1$^{hi}$ T$_{FH}$ cells were significantly reduced in *Orai1*$^{fl/fl}$*Orai2*$^{-/-}$*Cd4cre* mice compared to WT controls (Fig. 7a). In contrast, *Orai1*- or *Orai2*-deficient mice had normal numbers of T$_{FH}$ cells. Analysing CD19$^+$CD38$^-$GL.7$^+$ GC B cells by flow cytometry (Fig. 7b) and GCs within B-cell follicles using immunohistochemistry (Fig. 7c), we observed a strongly impaired GC reaction in *Orai1*$^{fl/fl}$*Orai2*$^{-/-}$*Cd4cre* but not single knockout or WT mice (Fig. 7b,c). Consequently, LCMV-specific, class-switched immunoglobulin-G (IgG) antibodies were markedly decreased in *Orai1*$^{fl/fl}$*Orai2*$^{-/-}$*Cd4cre* (but not single knockout) mice, whereas LCMV-specific IgM antibodies were normal (Fig. 7d). Taken together, our data demonstrate that both ORAI1 and ORAI2 are required for T$_{FH}$ cell-dependent humoral immunity.

**Deletion of *Orai1* and *Orai2* prevents T cell-induced colitis.** T cells provide immunity to infection but also mediate inflammation in a variety of autoimmune diseases[47,48]. To elucidate the role of *Orai1* and *Orai2* in autoimmunity, we used an adoptive transfer model of IBD that tests the function of ORAI1 and ORAI2 in pathogenic donor T cells[49]. We injected naive CD4$^+$ T cells from WT, *Orai1*$^{fl/fl}$*Cd4cre*, *Orai2*$^{-/-}$ and *Orai1*$^{fl/fl}$ *Orai2*$^{-/-}$*Cd4cre* mice into lymphopenic *Rag1*$^{-/-}$ mice to

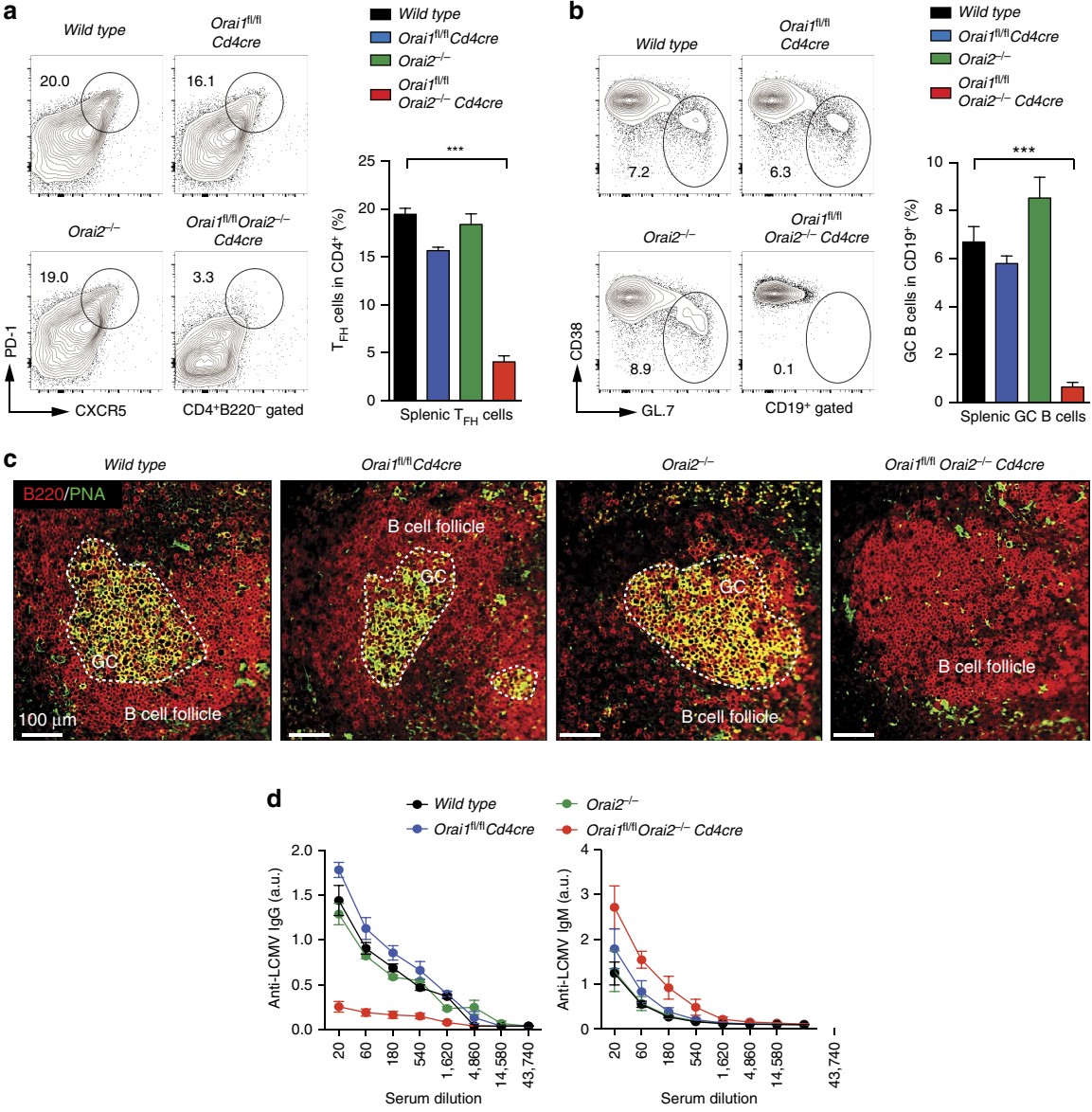

**Figure 7 | Humoral immunity requires both ORAI1 and ORAI2 in T$_{FH}$ cells. (a,b)** Analysis of CD4$^+$CXCR5$^{hi}$PD-1$^{hi}$ T$_{FH}$ cells (**a**) and CD19$^+$CD38$^-$GL.7$^+$ GC B cells (**b**) in spleens of WT, *Orai1$^{fl/fl}$Cd4cre*, *Orai2$^{-/-}$* and *Orai1$^{fl/fl}$Orai2$^{-/-}$Cd4cre* (DKO) mice 10 days after LCMV infection by flow cytometry; means ± s.e.m. of four to eight mice. (**c**) Representative immunofluorescence of GCs in WT, *Orai1$^{fl/fl}$Cd4cre*, *Orai2$^{-/-}$* and DKO mice 10 days after LCMV infection; PNA (green) and B220 (red). Scale bar, 100 μm. (**d**) Analysis of LCMV-specific IgM and IgG levels in the sera of WT, *Orai1$^{fl/fl}$Cd4cre*, *Orai2$^{-/-}$* and DKO mice 10 days after LCMV infection; means ± s.e.m. of four to six mice. *$P<0.05$; **$P<0.005$; ***$P<0.001$ in (**a,b**) using unpaired Student's *t*-tests.

induce IBD. Mice that had received T cells from WT or *Orai1$^{fl/fl}$ Cd4cre* mice lost ~10% of their original body weight (Fig. 8a). Transfer of *Orai2*-deficient T cells caused more pronounced (but statistically not significant) weight loss compared to WT cells. Importantly, T cells from *Orai1$^{fl/fl}$Orai2$^{-/-}$Cd4cre* mice did not induce any weight loss (Fig. 8a). Macroscopic inspection of colons after 12 weeks showed colon shortening, thickening and bleeding after transfer of WT or *Orai2*-deficient T cells, and to a lesser extent of *Orai1*-deficient T cells (Fig. 8b). In contrast, colons appeared healthy in mice that were injected with *Orai1$^{fl/fl}$ Orai2$^{-/-}$Cd4cre* T cells (Fig. 8b). The histopathological examination of colonic sections after transfer of *Orai1/Orai2*-deficient T cells revealed no IBD symptoms (such as immune cell infiltration, epithelial hyperplasia, goblet cell depletion and/or ulceration) resulting in low IBD scores of <1 (Fig. 8c,d). In contrast, mice with T cells from WT, *Orai1$^{fl/fl}$Cd4cre* or *Orai2$^{-/-}$* mice had marked colon inflammation (Fig. 8c,d).

Consistent with reduced colitis, the frequency and absolute numbers of CD4$^+$ T cells in the spleen, mesenteric LNs and lamina propria (LP) were significantly reduced in mice that had received T cells from *Orai1$^{fl/fl}$Orai2$^{-/-}$Cd4cre* mice compared to WT or single knockout mice (Fig. 8e and Supplementary Fig. 7). Proinflammatory cytokines including IFNγ, tumour-necrosis factor-α (TNFα) and IL-17, which are produced by colitogenic T$_H$1 and T$_H$17 cells, are critically involved in IBD pathogenesis. *Orai1*- and *Orai1/Orai2*-deficient donor T cells isolated from the spleen, mesenteric LNs and LP had strongly impaired production of IFNγ, TNFα and IL-17 after restimulation *ex vivo* compared to WT or *Orai2*-deficient T cells (Fig. 8f). It is noteworthy that although cytokine production was similarly impaired in *Orai1*- and *Orai1/Orai2*-deficient T cells (Fig. 8f), only mice that had received *Orai1/Orai2*-deficient T cells were fully protected from colitis, which is likely explained by the combination of abrogated cytokine production, proliferation and

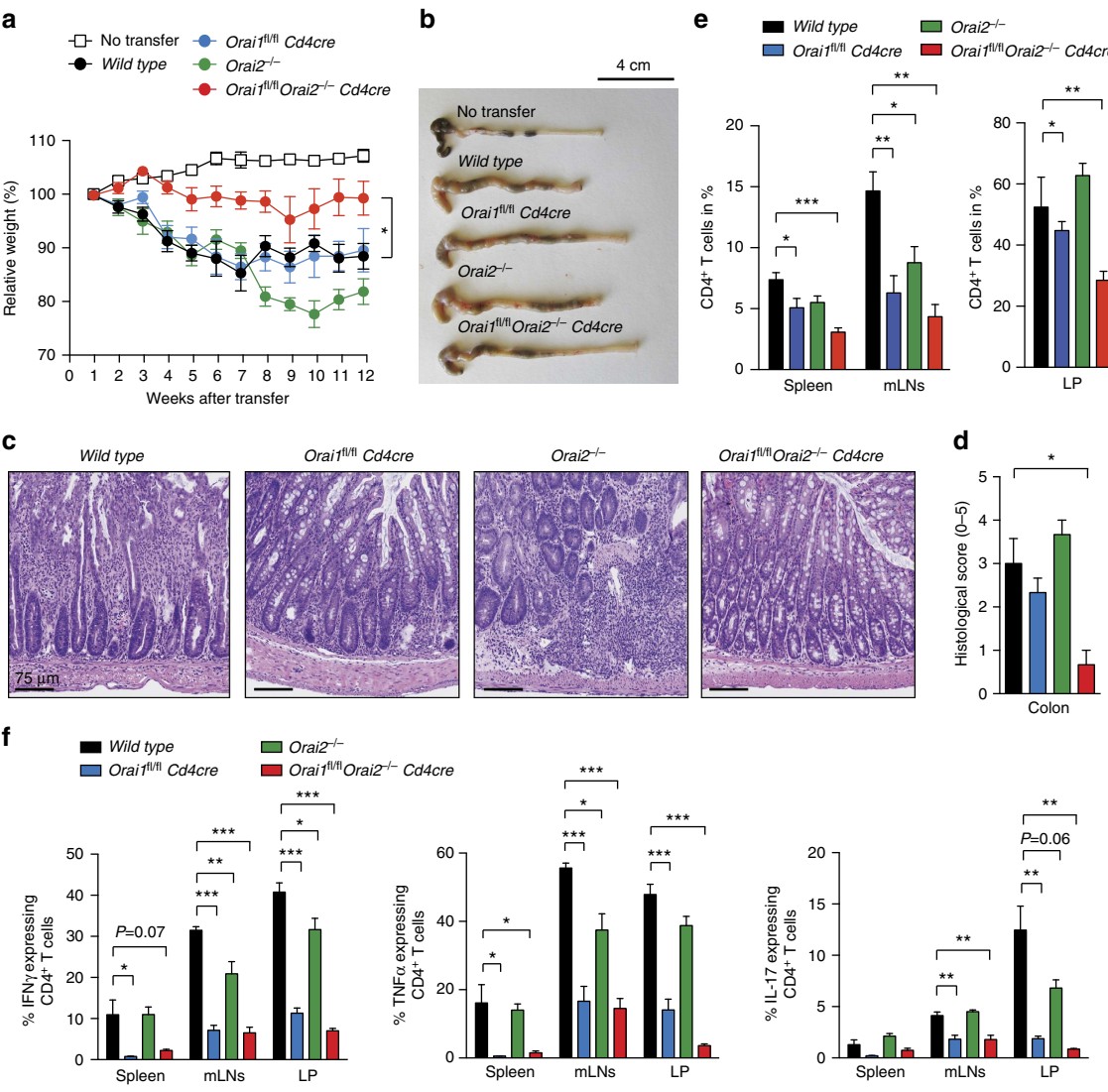

**Figure 8 | *Orai1/Orai2*-deficient T cells fail to induce colitis after adoptive transfer into lymphopenic host mice.** (**a**) Weight loss of *Rag1*$^{-/-}$ host mice after transfer of $5 \times 10^5$ CD4$^+$CD25$^-$CD62L$^{hi}$ naive T cells from WT, *Orai1*$^{fl/fl}$*Cd4cre*, *Orai2*$^{-/-}$ and *Orai1*$^{fl/fl}$*Orai2*$^{-/-}$*Cd4cre* (DKO) mice; means ± s.e.m. of 9–13 mice. (**b**) Representative macroscopic pictures of colon samples from mice described in **a**. Scale bar, 4 cm. (**c**) Representative H&E colon histologies 12 weeks after naive T cell transfer. Scale bar 75 μm. (**d**) Histopathological examination of H&E-stained colon sections as shown in **c**; means ± s.e.m. of three to four mice. (**e**) Frequency of CD4$^+$ T cells in spleen, mesenteric LNs (mLNs) and LP of *Rag1*$^{-/-}$ mice 12 weeks after naive T cell transfer; means ± s.e.m. of four to five mice. (**f**) Cytokine production by T cells isolated from spleen, mLNs and lamina propria (LP) 12 weeks after naive T cell transfer. Intracellular IFNγ, TNFα and IL-17 levels were measured after stimulation with phorbol myristate acetate (PMA)/ionomycin and analysed by flow cytometry; means ± s.e.m. of four to five mice. *$P < 0.05$; **$P < 0.005$; ***$P < 0.001$ using unpaired Student's *t*-test in (**e**) and two-way analysis of variance (ANOVA) in **f**.

decreased tissue homing, all of which were more pronounced in *Orai1/Orai2*-deficient compared to *Orai1*-deficient T cells.

**Deletion of *Orai1* and *Orai2* in T cells prevents GvHD**. GvHD is a complication of BM transplantation during which allogenic T cells in the transplant cause inflammation of the recipient's liver, skin and gastrointestinal tract[50]. To investigate the role of ORAI1 and ORAI2 in donor T cells after allogenic hematopoietic cell transplantation (allo-HCT), we transplanted total T cells from either WT, *Orai1*$^{fl/fl}$*Cd4cre*, *Orai2*$^{-/-}$ or *Orai1*$^{fl/fl}$*Orai2*$^{-/-}$*Cd4cre* C57BL/6 mice, together with BM cells from WT C57BL/6 mice into lethally irradiated allogenic BALB/c host mice. Transplantation of WT or *Orai2*$^{-/-}$ T cells caused fulminant GvHD (using a composite clinical GvHD score as described in ref. 51) after allo-HCT, whereas *Orai1*$^{fl/fl}$*Cd4cre* T cells resulted in markedly reduced immunopathology (Fig. 9a,b). Strikingly,

*Orai1/Orai2*-deficient T cells did not cause any clinical GvHD symptoms including weight loss (Fig. 9a,b) and were indistinguishable from control mice that had been transplanted with BM from C57BL/6 mice only.

In acute GvHD, levels of IFNγ and TNFα correlate with disease activity[50,51]. IFNγ and TNFα levels in the sera of recipient mice (Fig. 9c) as well as IFNγ and TNFα production by donor CD4$^+$ and CD8$^+$ T cells restimulated *ex vivo* (Fig. 9d) were strongly impaired in mice that had received *Orai1*-deficient T cells or *Orai1*$^{fl/fl}$*Orai2*$^{-/-}$*Cd4cre* T cells (Fig. 9c,d). In contrast, transfer of *Orai2*$^{-/-}$ or WT T cells resulted in comparable and robust cytokine production. The histopathological analysis of liver, small intestine and large intestine showed pronounced immune cell infiltration and inflammation in mice transplanted with WT or *Orai2*$^{-/-}$ T cells, whereas less or no immunopathology was apparent in recipients of *Orai1*- or *Orai1/Orai2*-deficient T cells, respectively (Fig. 9e). Although mice that had received

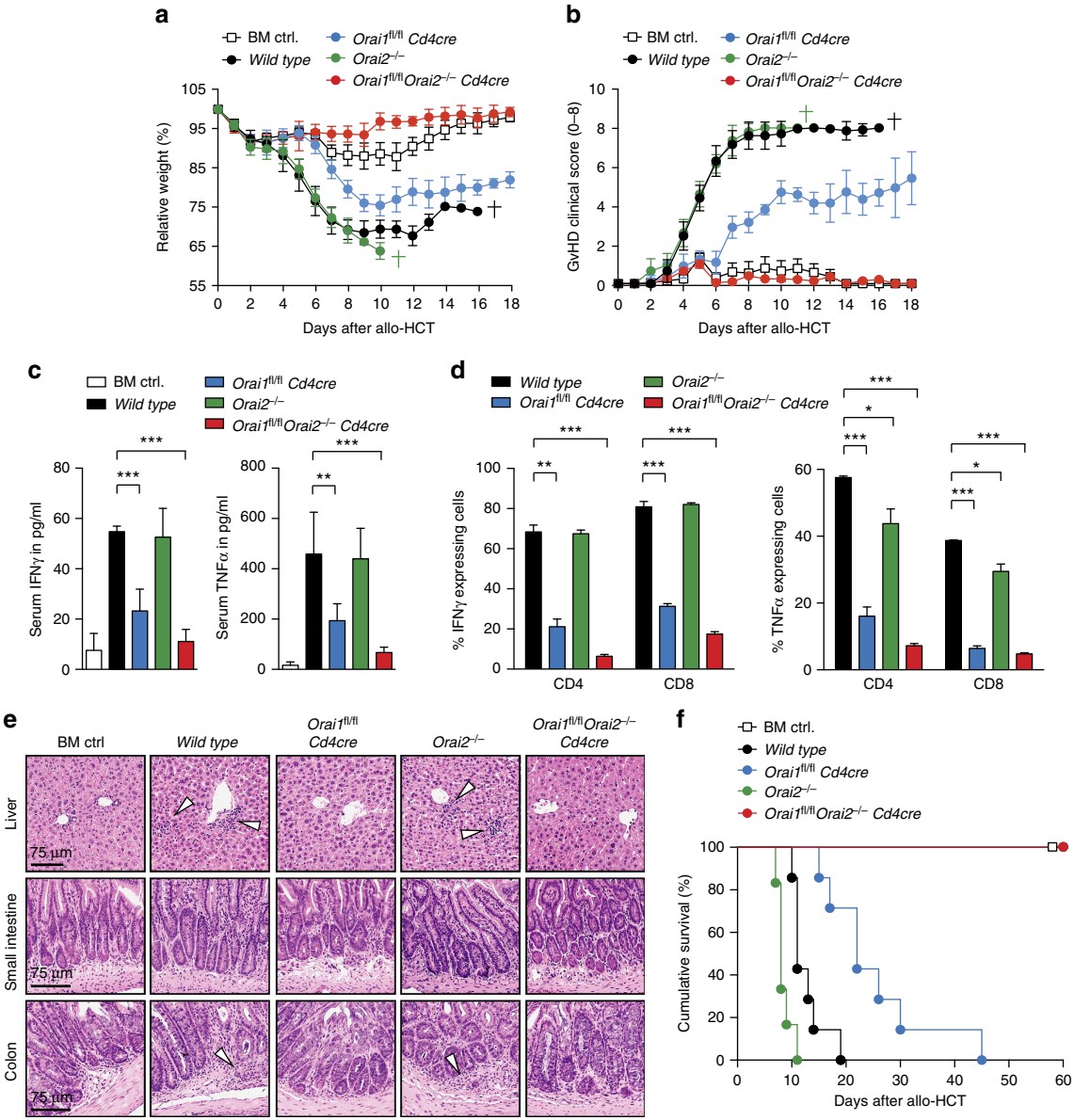

**Figure 9 | *Orai1/Orai2*-deficient T cells fail to cause GvHD after adoptive transfer into allogenic host mice.** (**a**) Weight loss of BALB/c host mice after transfer of allogenic T cells from WT, *Orai1^{fl/fl}Cd4cre*, *Orai2^{-/-}* and *Orai1^{fl/fl}Orai2^{-/-} Cd4cre* (DKO) donor mice. BALB/c mice were lethally irradiated with 8 Gy and transplanted with $5 \times 10^6$ allogenic C57BL/6 WT BM cells together with $1.2 \times 10^6$ allogenic C57BL/6 T cells from WT, *Orai1^{fl/fl} Cd4cre*, *Orai2^{-/-}* and DKO mice. Mice without T cell transfer (BM ctrl.) were used as control; means ± s.e.m. of seven mice. (**b**) Clinical scores of GvHD pathology (0–8) after allogenic T cell transfer described in **a**; means ± s.e.m. of seven mice. (**c**) Serum concentration of IFNγ and TNFα in host mice 10 days after allogenic T cell transfer as described in **a** analysed by ELISA; means ± s.e.m. of four to five mice. (**d**) Analysis of intracellular IFNγ and TNFα in WT, *Orai1^{fl/fl} Cd4cre*, *Orai2^{-/-}* and DKO T cells 10 days after transfer and following stimulation with phorbol myristate acetate (PMA)/iono; means ± s.e.m. of three mice. (**e**) Representative H&E-stained microsections of the liver, small intestine and colon of BALB/c host mice 10 days after allogenic transfer of WT, *Orai1^{fl/fl}Cd4cre*, *Orai2^{-/-}* and DKO T cells. (**f**) Cumulative survival of BALB/c host mice after allogenic transfer of WT (mean survival time (MST): 11 days), *Orai1^{fl/fl}Cd4cre* (MST: 22 days), *Orai2^{-/-}* (MST: 8 days) and *Orai1^{fl/fl}Orai2^{-/-} Cd4cre* T cells (MST: >60 days); seven mice per group as described in **a**. $^*P<0.05$; $^{**}P<0.005$; $^{***}P<0.001$ in (**c,d**) using two-way analysis of variance (ANOVA).

*Orai1*-deficient T cells survived significantly longer than recipients of WT or *Orai2^{-/-}* T cells, only animals transplanted with *Orai1/Orai2*-deficient T cells were completely protected from lethal GvHD (Fig. 9f). Taken together, loss of *Orai2* moderately exacerbated and loss of *Orai1* ameliorated lethal GvHD, whereas combined deletion of both *Orai1* and *Orai2* in T cells fully protected recipient mice from GvHD, demonstrating that ORAI1 and ORAI2 together regulate pathogenic T cell function.

## Discussion

The importance of ORAI1 for SOCE and immune function is well established[17,44,45], but that of ORAI2 and ORAI3 remains essentially unknown. Here we find that ORAI1 and ORAI2 are highly expressed in mouse T cells. Deletion of *Orai1* moderately decreased SOCE in naive and more strongly in effector T cells. By contrast, deletion of *Orai2* enhanced SOCE in naive but not effector T cells. Combined deletion of *Orai1* and *Orai2* completely abolished SOCE, providing strong evidence that both

ORAI1 and ORAI2 mediate SOCE in T cells, whereas ORAI3 is dispensable. We here show that ORAI1 and ORAI2 form a heteromeric channel complex, in which ORAI2 attenuates the function of ORAI1 and limits SOCE. This effect is at least in part due to smaller steady-state CRAC currents in cells coexpressing ORAI1 and ORAI2, likely caused by enhanced fast CDI in these cells. Abolishing SOCE by combined deletion of $Orai1$ and $Orai2$ resulted in severe defects in T cell function $in$ $vitro$ and T cell-mediated immune responses $in$ $vivo$ including antibody production after viral infection, colitis and GvHD. Individual deletion of ORAI1 or ORAI2 had less pronounced or no effects, respectively.

The inhibitory role of ORAI2 in CRAC channel function is not limited to naive T cells. We also observed increased SOCE in $Orai2$-deficient neutrophils, astrocytes, dendritic cells and macrophages as well as shORAI2-treated human fibroblasts. A similar enhancement of SOCE has recently been reported in Jurkat T cells and a chondrocyte cell line after knockdown of ORAI2 by RNA interference[52,53]. These data indicate that ORAI2 attenuates CRAC channel function and SOCE in many cell types. We show that increased SOCE in $Orai2$-deficient cells is due to increased $I_{CRAC}$ mediated by ORAI1. CRAC currents were increased in $Orai2$-deficient BMDMs and had biophysical properties comparable to native CRAC currents. Furthermore, Fura-2 quench rates by extracellular $Mn^{2+}$ were larger in $Orai2$-deficient BMDMs and T cells. Enhanced SOCE in ORAI2-deficient cells was not due to greater expression of ORAI1 or altered $Ca^{2+}$ content of ER stores. ORAI2 has been proposed to be an ER release channel that controls the $Ca^{2+}$ filling state of the ER and may thereby control SOCE[33]. If this was the main function of ORAI2, we would have expected that ORAI2-deficient cells have more $Ca^{2+}$ stored in their ER, which was, however, not the case.

Our finding that combined deletion of ORAI1 and ORAI2 abolishes $I_{CRAC}$ and SOCE, whereas single deletion of ORAI2 enhances both is best explained by a model in which ORAI1 and ORAI2 form heteromeric channels. CRAC channels are formed by multimers of ORAI proteins, which most likely form hexameric complexes[1,2]. While most studies have investigated homomeric ORAI channels by ectopic expression of either ORAI1, ORAI2 or ORAI3, it is likely that CRAC channels are heteromers of different ORAI homologues. This idea is supported by coexpression of different ORAI homologues in many tissues[14] as well as co-immunoprecipitation[14] and fluorescence resonance energy transfer[53] between ectopically expressed ORAI1 and ORAI2 proteins. We here demonstrate that ORAI1 and ORAI2 together form functional heteromeric CRAC channels in naive T cells (and other cell types) in which both ORAI1 and ORAI2 subunits form the channel pore. Previous studies have revealed that ectopically expressed ORAI2 generates CRAC currents with similar properties but smaller amplitudes than ORAI1 (refs 8,9). Thus, deletion of ORAI2 protein from heteromeric CRAC channels in naive T cells might be expected to produce homomeric ORAI1 channels with larger currents resulting in increased SOCE, which is consistent with our findings. Although the presence of ORAI2 in heteromeric ORAI1:ORAI2 channels has an inhibitory effect on CRAC channel function and SOCE, ORAI2 is a functional CRAC channel. This is apparent from the fact that residual SOCE in T cells from $Orai1^{fl/fl}Cd4cre$ mice is abolished when $Orai2$ is also deleted and that overexpression of ORAI2 results in currents resembling those of endogenous CRAC channels consistent with published reports[8,9].

Evidence supporting direct interaction of ORAI1 and ORAI2 in heteromeric complexes comes from our experiments in which expression of the pore-dead ORAI1$^{E106Q}$ mutant in $Orai1$-deficient T cells (whose residual SOCE is mediated by ORAI2)

abolished SOCE. Conversely, expression of the pore-dead ORAI2$^{E80Q}$ mutant in $Orai2$-deficient T cells (whose enhanced SOCE is mediated by ORAI1) also abolished SOCE. These effects of ectopically expressed pore-dead ORAI mutants on SOCE are unlikely due to their sequestration of STIM1. Rather, we conclude that pore-dead ORAI proteins directly assemble with their endogenous homologues into heteromeric channels and abolish $Ca^{2+}$ conductance. This conclusion is supported by experiments in which we overexpressed ORAI1, ORAI2 or both ORAI1/ORAI2 together with STIM1. Whereas coexpression of WT ORAI1 and ORAI2 results in very large CRAC currents, coexpression of the pore-mutant ORAI1$^{E106A}$ together with ORAI2 abolished $I_{CRAC}$. Since STIM1 is abundant under these conditions, the effects of ORAI1$^{E106A}$ are not due to STIM1 sequestration away from ORAI2 but instead support the model of direct ORAI1/ORAI2 interaction in a heteromeric channel.

Enhanced $I_{CRAC}$ and SOCE in ORAI2-deficient cells and evidence for heteromeric ORAI1:ORAI2 channels suggest that ORAI2 is less efficient than ORAI1 or ORAI1:ORAI2 channels in conducting $Ca^{2+}$. It is therefore noteworthy that peak current amplitudes were comparable in cells expressing ORAI1, ORAI2 or ORAI1/ORAI2, indicating that homomeric and heteromeric ORAI channels have similar abilities to conduct $Ca^{2+}$. By contrast, steady-state currents were significantly larger in cells expressing ORAI1 compared to ORAI2, which can be explained by the more pronounced fast CDI observed in cells expressing ORAI2 compared to ORAI1 consistent with previous reports[9,38]. More studies are needed to elucidate the mechanistic differences in fast CDI and ion conduction between ORAI1 and ORAI2. An alternative explanation for enhanced SOCE in $Orai2^{-/-}$ T cells that merits further investigation is that gating of ORAI1 channels by STIM1 is more efficient than that of ORAI2 channels[38,54].

The concept of heteromeric CRAC channel complexes is intriguing because the dynamic variation of the stoichiometry of two homologues with higher (ORAI1) and lower (ORAI2) $Ca^{2+}$ permeation provides a mechanism to regulate SOCE while maintaining essential CRAC channel properties. We found that in naive T cells both ORAI1 and ORAI2 contribute to CRAC channel function with ORAI2 assuming an inhibitory role apparent from increased SOCE in naive $Orai2^{-/-}$ T cells. In effector T cells, ORAI1 was up- and ORAI2 downregulated compared to naive T cells, resulting in an increased ORAI1:ORAI2 expression ratio (Supplementary Fig. 8). As a consequence, deletion of $Orai2$ had little effect on SOCE in effector T cells, whereas deletion of $Orai1$ suppressed SOCE more strongly than in naive T cells. This reciprocal expression of ORAI1 and ORAI2 in naive versus effector T cells provides a mechanism to fine-tune the magnitude of SOCE and thus the strength of T cell-mediated immune responses. From an immunological perspective, increasing the number of ORAI2 subunits in heteromeric CRAC channels represents a means to reduce the magnitude of SOCE and restrain the strength of TCR signalling, thus reducing the likelihood of random, detrimental activation of naive T cells. By contrast, effector T cells have to respond quickly and effectively to cognate antigens. Increasing the ORAI1:ORAI2 ratio and thus the magnitude of SOCE ensures robust T cell activation and efficient recall responses.

We here show that ORAI1 and ORAI2 form the CRAC channel complex in T cells and are required for T cell-mediated immune responses, whereas ORAI3 is dispensable. T cells from $Orai1^{fl/fl}Orai2^{-/-}Cd4cre$ mice not only had abolished SOCE but also lacked proliferation, cytokine production, $T_{FH}$ cell differentiation and virus-specific antibody production. Furthermore, $Orai1/Orai2$-deficient T cells did not cause immunopathology in adoptive transfer models of colitis and GvHD. These findings are reminiscent of $Stim1^{fl/fl}Stim2^{fl/fl}Cd4cre$ mice, whose T cells lack

SOCE and fail to differentiate into $T_{FH}$ cells in response to viral infection or immunization[18]. Likewise, $Orai1^{fl/fl}Orai2^{-/-}$ Cd4cre mice had impaired $T_{FH}$ cell differentiation, GC formation and antibody production. Individual deletion of Orai1 or Orai2 resulting in partially impaired or increased SOCE, respectively, had no effect on T cell-proliferation in vitro or $T_{FH}$ cell differentiation, GC formation and antibody production in vivo. Thus, only complete elimination of SOCE in Orai1/Orai2- or Stim1/Stim2-deficient mice compromises proliferation and T cell-dependent antibody production. Furthermore, $Orai1^{fl/fl}Orai2^{-/-}$ Cd4cre mice had reduced Treg numbers[18,39,40] and accumulation of effector/memory T cells in secondary lymphoid organs, which caused splenomegaly and lymphadenopathy, that is also observed in $Stim1^{fl/fl}$ $Stim2^{fl/fl}$Cd4cre mice[39].

Enhanced SOCE in naive $Orai2^{-/-}$ T cells might have been expected to increase T cell function in vitro and T cell-dependent immunopathologies in vivo, which was however not the case. The severity of IBD was only moderately increased after adoptive transfer of naive $Orai2^{-/-}$ T cells compared to WT controls. Likewise, GvHD after transfer of Orai2-deficient T cells was not significantly more severe compared to that induced by WT cells. A likely explanation for this lack of exacerbation of immune responses by Orai2-deficient T cells is the lower expression of ORAI2 in effector T cells compared to naive T cells and the resulting predominance of ORAI1 in effector T cells. Indeed, deletion of Orai2 increased SOCE in naive but not effector T cells. Since immunopathology in the IBD and GvHD models is mediated by effector T cells that have normal SOCE in the absence of ORAI2, exacerbated immunopathology would not be expected in $Orai2^{-/-}$ mice. In human T cells, ORAI2 is expressed albeit at lower levels than in murine T cells. Although somatic mutations of ORAI2 were reported in cancer[55], no functional ORAI2 germline mutations have been reported. By contrast, null or loss-of-function mutations in the human ORAI1 gene that abolish SOCE severely impaired T cell function[17,44]. Patients with inherited mutations in ORAI1 present clinically with CRAC channelopathy characterized by a severe combined immunodeficiency-like disease, autoimmunity, anhidrotic ectodermal dysplasia and congenital myopathy[17,44,45]. Although abolished SOCE, T cell dysfunction and immunodeficiency in these patients indicate that ORAI1 is the dominant CRAC channel subunit in human effector T cells, these findings do not exclude a potential role of ORAI2 in other immune or non-immune cells in which it is expressed.

## Methods

**Mice.** $Orai1^{fl/fl}$Cd4cre mice were described before[25]. $Rag1^{-/-}$ mice (JAX strain number 002216) and BALB/c WT mice (JAX strain number 000651) were purchased from The Jackson Laboratory. $Orai2^{LacZ/+}$ mice carrying a LacZ reporter cassette were generated using VGB6 embryonic stem cells (C57BL/6NTac) obtained from the Knock-Out Mouse Project (KOMP) repository at UC Davis (project ID VG14962, Orai2tm1(KOMP)Vlcg). The targeting strategy for generating $Orai2^{LacZ/+}$ mice is shown in Supplementary Fig. 2a. Exon 4 and the 5′ region of exon 5 representing the coding sequence of Orai2 were replaced by homologous recombination with a loxP-flanked neomycin gene (neo$^R$) and a LacZ reporter gene. The neo$^R$ gene was later removed by mating $Orai2^{LacZ/+}$ offsprings to CMV-Cre mice, followed by removal of the Cre transgene by backcrossing $Orai2^{LacZ/+}$ mice to C57BL/6 mice. The resulting offspring was intercrossed to generate $Orai2^{-/-}$ ($Orai2^{LacZ/LacZ}$) mice. Genotyping of mice for the presence of the WT allele, inserted neo$^R$ gene and deleted $Orai2^{-/-}$ allele was performed using mouse tail DNA and PCR (as shown in Supplementary Fig. 2b) using the following primer pairs: WT allele 5′-CTCGGCAGTTGCCTGTTTG-3′ and 5′-ACAGCCACCACGCTCATC-3′ (product ~89 bp); $Orai2^{-/-}$ allele 5′-GGTAAACTGGCTCGGATTAGGG-3′ and 5′-TTGACTGTAGCGGCTGATG TTG-3′ (product ~210 bp); neo$^R$ gene 5′-TCATTCTCAGTATTGTTTT GCC-3′ and 5′-CTTTGGGAATTCACCACCC-3′ (product ~400 bp). PCR conditions for all reactions were the following: denaturation at 95 °C for 10 min, followed by 40 cycles of 95 °C for 60 s, annealing at 55 °C for 30 s and elongation at 72 °C for 30 s with an additional 10 min final elongation step at 72 °C. All animals

were on a C57BL/6 genetic background. Both male and female mice were used between 6 and 16 weeks of age (unless otherwise noted). Sample size for various animal experiments was chosen based on prior data generated in the laboratory, no mice were excluded from experiments. Mice were maintained under specific pathogen-free conditions. All animal experiments were approved by the Institutional Animal Care and Use Committee at New York University School of Medicine.

**Plasmids.** Murine stem cell virus (MSCV)-based retroviral vectors expressing ORAI1$^{WT}$, ORAI2$^{WT}$ and ORAI1$^{E106Q}$ were described before[5,14,20]. ORAI2$^{E80Q}$, ORAI1$^{L273D}$ and ORAI1$^{L273D/E106Q}$ retroviral vectors were generated using the QuikChange II XL Site-directed Mutagenesis Kit (Agilent; cat. no. 200521) according to the manufacturer's instructions. Mutations were confirmed by DNA sequencing. Expression plasmids used in patch-clamp experiments were as follows: N-terminally YFP-tagged ORAI1 was purchased from GeneCopoeia. N-terminally CFP-tagged ORAI2 was described before[5,37]. The ORAI1$^{E106A}$ mutation was generated by site-directed mutagenesis using the QuickChange II XL Site-directed Mutagenesis Kit (Agilent). The indicated ORAI1 or ORAI2 constructs were transfected into HEK293 cells together with a construct expressing unlabelled STIM1 (pCMV6-XL5; OriGene Technologies) using Lipofectamine (Thermo Fisher; cat. no. L3000015). The primer sequences for site-directed mutagenesis are listed in Supplementary Table 3.

**Cells.** Human Hs27 fibroblasts (ATCC) were cultured in RPMI-1640 medium supplemented with 10% FCS (HyClone), and grown at 37 °C, 5% $CO_2$. HEK293H cells (Thermo Fisher Scientific, Cat.# 11631017) for electrophysiological recordings were maintained in suspension in a medium containing CD293 supplemented with 4 mM GlutaMAX (Invitrogen) at 37 °C, 5% $CO_2$. For electrophysiological analysis of ORAI1 and/or ORAI2 currents, the cells were plated onto poly-L-lysine-coated coverslips at the time of passage, and grown in a medium containing 44% DMEM (Mediatech), 44% Ham's F12 (Mediatech), 10% FCS (HyClone), 2 mM glutamine, 50 U ml$^{-1}$ penicillin and 50 µg ml$^{-1}$ streptomycin (Invitrogen).

**Generation of BMDMs.** Femurs and tibiae of mice were removed and BM was flushed out using a 0.45-mm-diameter needle and washed two times. $5 \times 10^6$ BM cells were cultured in 10 ml DMEM containing 10% CMG14-12 cell supernatant. After 6 days, BMDMs were >90% CD11b$^+$F4/80$^+$ as determined by flow cytometry[56].

**T cell stimulation and differentiation.** Total and CD4$^+$ T cells were isolated from the spleens and LNs using the Mouse pan T Cell Enrichment Kit or the Mouse CD4$^+$ T Cell Enrichment Kit, respectively (both STEMCELL Technologies; cat. nos 19852 and 19751). CD4$^+$ T cells were stimulated with 1 µg ml$^{-1}$ plate-bound anti-CD3 (clone 2C11) plus 1 µg ml$^{-1}$ anti-CD28 antibodies (clone 37.51; both Bio X Cell). For differentiation of naive CD4$^+$ T cells into $T_H1$, $T_H17$ and $T_H2$ cells, T cells were polarized for 3 days with 10 ng ml$^{-1}$ IL-12 (PeproTech) and 2 µg ml$^{-1}$ anti-IL-4 (eBioscience) for $T_H1$; 20 ng ml$^{-1}$ IL-6 (PeproTech), 0.5 ng ml$^{-1}$ hTGFβ1 (PeproTech), 2 µg ml$^{-1}$ anti-IL-4 (clone 11B11) and 2 µg ml$^{-1}$ anti-IFNγ (clone XMG1.2; both eBioscience) for $T_H17$ and 100 ng ml$^{-1}$ IL-4 (PeproTech), 5 µg ml$^{-1}$ anti-IL-12 and 20 µg ml$^{-1}$ anti-IFNγ (both eBioscience) for $T_H2$ cells in IMDM or RPMI medium (both Cellgro). Both media contained 2 mM L-glutamine, 50 mM 2-ME, 100 U ml$^{-1}$ penicillin/streptomycin and 10% FCS. For cytokine expression, cells were (re-)stimulated with 1 µM ionomycin plus 20 nM phorbol myristate acetate (both Calbiochem) for 6 h in the presence of brefeldin A (BioLegend) and analysed by flow cytometry as described below.

**Retroviral transduction of T cells.** CD4$^+$ T cells were isolated using a mouse CD4$^+$ T Cell Enrichment Kit (STEMCELL Technologies) and stimulated with 1 µg ml$^{-1}$ plate-bound anti-CD3 (clone 2C11) plus 1 µg ml$^{-1}$ anti-CD28 antibodies (clone 37.51; both Bio X Cell) in the presence of 50 U ml$^{-1}$ IL-2 (Peprotech). T cells were transduced 24 h after stimulation by spin infection (2,500 r.p.m., 30 °C, 90 min) in the presence of concentrated retroviral supernatant and 10 µg ml$^{-1}$ polybrene (Santa Cruz Biotechnology). Retroviral supernatant was produced in the Platinum-E retroviral packaging cell line. Platinum-E cells (ATCC) were transfected with GenJet transfection reagent (SignaGen; cat. no. SL100499) with murine stem cell virus-based retroviral vectors expressing ORAI1$^{WT}$, ORAI2$^{WT}$ and ORAI1$^{E106Q}$ (as described in refs 5,14) as well as ORAI2$^{E80Q}$, ORAI1$^{L273D}$ and ORAI1$^{L273D/E106Q}$ together with the amphotrophic packaging vector pCL-10A1. Two days after transfection, the supernatant was collected and concentrated using Amicon Ultra-15 centrifugal filters (Merck Millipore; cat. no. UFC910024). At 4 h after spin infection, viral supernatant was removed, and then replaced by fresh media. At 48 h after transduction, T cells were transferred into new plates adding fresh media containing 50 U ml$^{-1}$ IL-2. Transduced cells were FACS sorted using a sterile Sony SY3200 (HAPS1) cell sorter and $Ca^{2+}$ influx was tested using a FlexStation 3 plate reader.

**Flow cytometry and cell sorting.** Cells isolated from spleen, LNs and LP were washed in cold PBS containing 0.1% BSA and unspecific binding was blocked using anti-FcγRII/FcγRIII (2.4G2; eBioscience). Staining of surface molecules with fluorescently labelled antibodies was performed at 4 °C for 30 min in the dark. Intracellular cytokine (IC) staining was performed using the IC Staining Buffer Kit (eBioscience; cat. no. 00-8222-49) according to the manufacturer's instructions. Surface expression of human ORAI1 in fibroblasts was detected using two different custom-made anti-human ORAI1 monoclonal antibodies, which both recognize the second extracellular loop of hORAI1. Clone 29A2 is a mouse anti-human ORAI1 mAb (IgG1) that was generated with an ovalbumin conjugated peptide corresponding to amino acids 196 to 234 of human ORA1 using mice that over-express the neonatal Fc receptor (FcRn) and have augmented immune response due to the role of FcRn in antigen presentation[62]. Immune sera and hybridoma supernatants were tested using human ORAI1 transfected and parenteral NIH-3T3 cells with flow cytometry analyses. A mouse IgG1 Ab was used as isotype control and a goat anti-mouse IgG conjugated to AlexaFluor 647 (Life technologies) as secondary antibody for detection. Clone 2C1.1 is a fully human anti-ORAI1 monoclonal antibody conjugated to allophycocyanin (APC) as described in ref. 57. A complete list of antibodies with the respective conjugated fluorochromes can be found in Supplementary Table 1. Samples were acquired on an LSRII flow cytometer using FACSDiva Software (BD Biosciences) and further analysed with FlowJo Software (Tree Star). Sorting of various cell populations by flow cytometry was performed using a sterile Sony SY3200 (HAPS1) cell sorter. For detection of β-galactosidase activity in lacZ (Orai2 reporter) positive cells, cells were loaded with 2 mM of the green fluorogenic substrate fluorescein di-V-galactoside (FDG) (Sigma; cat. no. F2756) in Hank's balanced salt solution containing 2% FBS, 10 mM HEPES (pH 7.2) according to the manufacturer's instructions. After loading, cells were incubated for 1.5 h on ice, washed once and further stained with flow cytometry antibodies. Representative gating strategies for all experiments can be found in Supplementary Fig. 9.

**T cell proliferation assay.** A total of $5 \times 10^6$ T cells were loaded with carboxyfluorescein diacetate succinimidyl ester (CFSE) (Invitrogen; cat. no. C34554) according to the manufacturer's instructions. Cells were blocked with 50% FBS, washed twice with RPMI and stimulated with $1 \mu g \, ml^{-1}$ plate-bound anti-CD3 (clone 2C11) plus $1 \mu g \, ml^{-1}$ anti-CD28 antibodies (clone 37.51; both Bio X Cell) with or without $50 \, U \, ml^{-1}$ IL-2 (NIH) for 4 days. CFSE dilution was monitored daily using an LSRII flow cytometer and FACSDiva Software (BD Biosciences) and further analysed with FlowJo Software (Tree Star).

**Quantitative real-time PCR.** RNA was extracted from FACS-sorted T cells and BMDMs using the RNeasy Micro Kit (Qiagen; cat. no. 74004), followed by cDNA synthesis with the iScript II Kit (Bio-Rad; cat. no. 170-8890). qRT–PCR was performed using Platinum SYBR Green qPCR SuperMix (Invitrogen; cat. no. 11744100) and an Opticon 2 thermocycler (Bio-Rad). PCR conditions were as follows: 95 °C for 10 min, followed by 40 cycles (94 °C for 45 s, 58 °C for 30 s and 72 °C for 30 s) of amplification. For quantitation, $C_T$ values were normalized to *Hprt* and expression was analysed using the $2^{-\Delta\Delta CT}$ method. Additional housekeeping genes (*Actin*, *Nono*, *18S*) were used to validate *Orai1* and *Orai2* expression in T cells, yielding similar results compared with *Hprt* (not shown). A complete list of all primers used in this study can be found in Supplementary Table 2.

**shRNA-mediated gene knockdown.** HEK293T cells (ATCC) were transfected with SGEP (ref. 19) and pLKO.1 (Addgene) lentiviral plasmids expressing shRNAs specific for human *ORAI1* and *ORAI2*, respectively, or scrambled shRNA together with packaging plasmid psPAX2 (Addgene; 12260) and vesicular stomatitis virus-G envelope plasmid pMD2.G (Addgene; cat. no. 12259) using the GenJet transfection reagent (SignaGen; cat. no. SL100499). *ORAI1* and *ORAI2* shRNA targeting sequences were 5′-TGTCCTCTAAGAGAATAAGCAT-3′ and 5′-CATCTTCGTGGTCTTCACCATC-3′ (Sigma), respectively. Human Hs27 fibroblasts (ATCC) were spin infected with fresh lentiviral supernatants for 90 min at 2,500 r.p.m. At 72 h post-transfection, transduced cells were selected with $1 \mu g \, ml^{-1}$ puromycin for 5 days. The knockdown efficiency of *ORAI1* and *ORAI2* genes was examined by qRT–PCR. All cell lines were tested for mycoplasma.

**Intracellular $Ca^{2+}$ levels and $Mn^{2+}$ quenching.** Cells were labelled with $2 \mu M$ Fura-2-AM (Life Technologies; cat. no. F1201) for 30 min in RPMI medium as described earlier[56]. Cells were attached for 10 min to 96-well imaging plates (Fisher) coated with 0.01% poly-L-lysine (w v$^{-1}$) (Sigma-Aldrich) and washed two times with $Ca^{2+}$-free Ringer solution (155 mM NaCl, 4.5 mM KCl, 2 mM $CaCl_2$, 1 mM $MgCl_2$, 10 mM D-glucose and 5 mM Na-HEPES) solution. Changes in intracellular $Ca^{2+}$ concentration were analysed using a FlexStation 3 plate reader (Molecular Devices) at 340 and 380 nm excitation wavelengths. Cells were stimulated with $1 \mu M$ thapsigargin or $0.3 \mu M$ ionomycin (both EMD Millipore) in $Ca^{2+}$-free Ringer solution and SOCE was analysed after readdition of 1 mM $Ca^{2+}$ (final) Ringer solution (for primary T cells and BMDMs) or 20 mM (final) $Ca^{2+}$ Ringer solution (for human fibroblast cells). SOCE was quantified by the peak or the slope of the F340/380 ratio. ER store depletion was analysed by the integrated $Ca^{2+}$ signal (area under the curve, AUC) after thapsigargin or ionomycin

stimulation and before readdition of 1 mM $Ca^{2+}$ Ringer solution. For measurements of store-operated bivalent cation influx via CRAC channels by $Mn^{2+}$ quenching, BMDMs were loaded with Fura-2-AM, transferred to $Ca^{2+}$-free Ringer solution and stimulated with $0.3 \mu M$ ionomycin or $1 \mu M$ thapsigargin. Fura-2 fluorescence was measured at an emission wavelength of 360 nm before and after addition of 1 mM $Mn^{2+}$.

**Patch-clamp measurements.** Currents were recorded in the standard whole-cell configuration at room temperature using an Axopatch 200B amplifier (Molecular Devices) interfaced to an ITC-18 input/output board (Instrutech). Currents were filtered at 1 kHz with a 4-pole Bessel filter and sampled at 5 kHz. Stimulation, data acquisition and analysis were performed using routines developed on the Igor Pro platform by R.S. Lewis (Stanford University, Palo Alto, CA). Recording electrodes were 'pulled' from 100 ml pipettes, were coated with Sylgard and were fire polished to a final resistance of 2–5 MΩ. All data were corrected for the liquid-junction potential of the pipette solution relative to that of Ringer's solution in the bath ($-10$ mV) and for leak currents collected in 20 mM extracellular $Ca^{2+}$ plus $La^{3+}$. The standard extracellular Ringer's solution contained 130 mM NaCl, 4.5 mM KCl, 20 mM $CaCl_2$, 10 mM D-glucose and 5 mM HEPES (pH 7.4) with NaOH. The DVF Ringer's solution contained 150 mM NaCl, 10 mM HEDTA, 1 mM EDTA and 10 mM HEPES (pH 7.4) with NaOH. 10 mM TEA-Cl was added to all extracellular solutions to block voltage-gated $K^+$ channels. The standard internal solution contained 135 mM caesium aspartate, 8 mM $MgCl_2$, 8 mM BAPTA and 10 mM HEPES (pH 7.2) with CsOH. The holding potential was $+30$ mV. The standard voltage stimulus consisted of a 100-ms step to $-100$ mV, followed by a 100-ms ramp from $-100$ to $+100$ mV applied at 1 s intervals. $I_{CRAC}$ was typically activated by depleting ER $Ca^{2+}$ stores with $1 \mu M$ thapsigargin, or by passive dialysis of BAPTA into the cell as indicated in the figure legends. Analysis of current amplitudes was performed by measuring either the peak, or in some cases, the steady-state current (as indicated in the figure legends) during the $-100$ mV step pulse.

**LCMV infection and detection of LCMV-specific antibodies.** The LCMV Armstrong (LCMV$^{ARM}$) strain was kindly provided by R. Ahmed (Emory University, Atlanta, GA). Virus was grown in BHK-21 cells and titres (PFU) in the supernatant were determined as described before[58]. For acute viral infections, mice were injected intraperitoneally with $2 \times 10^5$ PFU of LCMV$^{ARM}$ and analysed 10 days after infection[18,58]. LCMV-specific antibodies in the sera of mice were measured as described before[18,58]. Briefly, lysates of LCMV-infected BHK-21 cells were used as substrate and LCMV-specific IgG and IgM binding was detected in serial dilutions of serum using AP-conjugated goat-anti-mouse IgG or IgM antibodies (both Southern Biotech). Enzyme-linked immunosorbent assays (ELISAs) were developed with pNPP substrate (Thermo Scientific; cat. no. 37621) and absorbance was analysed at 405 nm using a SpectraMax M5 microplate reader (Molecular Devices).

**Adoptive transfer colitis.** Colitis was induced in *Rag1$^{-/-}$* host mice by intra-peritoneal injection of $5 \times 10^5$ CD4$^+$CD62L$^+$CD25$^-$ naive T cells isolated from LNs and spleen of WT, *Orai1*, *Orai2* and *Orai1/Orai2*-deficient mice and sorted by flow cytometry using a SONY SY3200 cell sorter. Greater than 95% enrichment of CD4$^+$CD62L$^+$CD25$^-$ T cells was confirmed by flow cytometry. Injected *Rag1$^{-/-}$* mice were assessed two times a week for weight loss and other symptoms of distress over a course of 12 weeks[24,59,60]. Proximal and distal colon sections were fixed in 4% paraformaldehyde, paraffin embedded and stained with haematoxylin and eosin (H&E) using standard protocols. Colon histology was scored in a blinded fashion by two individuals using a grading system from 0 to 5: 0, no changes; 1, minimal scattered mucosal inflammatory cell infiltrates, with or without epithelial hyperplasia; 2, mild scattered to diffuse inflammatory cell infiltrates, sometimes extending into the submucosa and associated with erosions, with minimal to mild epithelial hyperplasia and minimal to mild mucin depletion from goblet cells; 3, mild-to-moderate inflammatory cell infiltrates that were sometimes transmural, often associated with ulceration, with moderate epithelial hyperplasia and mucin depletion; 4, marked inflammatory cell infiltrates that were often transmural and associated with ulceration, with marked epithelial hyperplasia and mucin depletion; 5, marked transmural inflammation with severe ulceration and loss of intestinal glands.

**Graft-versus-host disease.** BALB/c host mice (H-2$^d$) were conditioned by myeloablative total body irradiation with a split-dose dose of 8.0 Gy total using a Gammacell 40 Exactor System (Best Theratronics). At 2 h after total body irradiation, mice were injected retroorbitally with $5 \times 10^6$ C57BL/6 WT BM cells (H-2$^b$) with or without $1.2 \times 10^6$ C57BL/6 total T cells (H-2$^b$). Animals were treated with antibiotic drinking water (sulfamethoxazole and trimethoprim; Hi-Tech Pharmacal) for 7 days before and after allo-HCT to prevent unspecific infections. Transplanted mice were assessed daily for weight loss and clinical symptoms of GvHD such as posture (hunching), fur texture, skin integrity and overall activity that were scored and used to calculate a composite GvHD score (adapted from Cooke *et al.*[51,61]). Mice were killed when GvHD symptoms reached our end-point scoring criteria, analysed for organ histopathology and used for cell isolation.

**Enzyme-linked immunosorbent assays.** TNFα and IFNγ serum levels in BALB/c host mice 12 days after allo-HCT were measured using commercial ELISA Kits (eBioscience; cat. no. 88-7324-22 and 88-8314-22) and analysed at 450 nm using a SpectraMax M5 microplate reader (Molecular Devices) according to the protocols provided by the manufacturer.

**Histochemistry and immunofluorescence (IF).** Livers, small intestine and colon of mice were collected, processed and H&E stained[51,59]. Whole E17.5 embryos were washed with PBS before fixation with 4% paraformaldehyde for 15 min. Tissue was washed with PBS containing 0.02% NP-40 three times for 30 min. Specimens were stained with 1 mg ml$^{-1}$ X-gal (Sigma) in X-gal buffer (5 mM $K_3Fe(CN)_6$, 5 mM $K_4Fe(CN)_6$, 2 mM $MgCl_2$, 0.01% Na deoxycholate, 0.02% NP-40; all Sigma) overnight at 30 °C. After washing three times with PBS, samples were stored in PBS containing 0.1% PFA or embedded in Tissue-Tek (Thermo Scientific) for histological analyses. For immunohistochemistry, tissue specimens were fixed in 4% PFA for at least 24 h and embedded in paraffin. Heat-induced antigen retrieval of tissue sections was performed in 10 mM citric acid buffer (pH 6.0) for 20 min using a high-pressure cooker (Deni). Blocking for 1 h with Antibody Diluent (Dako) was followed by overnight incubation at 4 °C with the following primary antibodies: rat anti-B220 (1:400, clone RA3-8B2; eBioscience) and biotinylated PNA (1:100; Vector Laboratories). Detection of IF was carried out using the following antibodies: goat anti-rat IgG (H + L) AlexaFluor555 (1:800) and streptavidin AlexaFluor488 (1:800; both Molecular Probes). Slides were mounted using Prolong Gold Antifade Mounting Medium (Molecular Probes). Images were captured using a Nikon microscope Eclipse TE2000. Data were analysed with NIS Elements Software (Nikon) and further processed using Photoshop.

**Immunoblot analysis.** Cells were lysed in RIPA lysis buffer (containing 50 mM Tris, 150 mM NaCl, 1% NP-40, 0.5% sodium deoxycholate, 0.1% SDS, 1 mM EDTA, 1 mM PMSF, 1 mM vanadate and complete protease inhibitor cocktail (Sigma; cat. no. P8340)) and sonicated on ice for 10 min. For western blots, 50–100 μg of total protein was fractionated by SDS–polyacrylamide gel electrophoresis and transferred onto nitrocellulose membrane. Membranes were incubated with a monoclonal goat anti-mouse actin antibody (1:5,000, clone C4; Santa Cruz Biotechnology) and a custom-made polyclonal rabbit anti-ORAI2 antibody (1:1,000, batch 1004, described in ref. 15) that recognizes the C terminus of ORAI2. For detection, peroxidase-coupled secondary anti-mouse or anti-rabbit antibodies (both Sigma) and the enhanced chemiluminescence system (Thermo Scientific) were used.

**Statistical analyses.** All data are shown as mean ± s.e.m. of at least three independent experiments. No specific randomization or blinding protocols were used. Figures were prepared using GraphPad Prism 6, Adobe Photoshop and Illustrator CC2014 Software. Different groups were compared using a two-tailed paired or unpaired Student's t-test or two-way analysis of variance using Prism6 (GraphPad Software). Differences with a P value of <0.05 were considered significant: *$P<0.05$; **$P<0.005$; ***$P<0.001$.

**Data availability.** The authors declare that all data supporting the findings of this study can be found within the paper and its Supplementary Information files. Additional data supporting the findings of this study are available from the corresponding author (S.F.) upon reasonable request.

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

## Acknowledgements

This work was funded by NIH Grants AI097302 (to S.F.) and GM114210 and NS057499 (to M.P.), postdoctoral fellowships VA 882/1-1 (to M.V.) and KA 4083/2-1 (to U.K.) by the Deutsche Forschungsgemeinschaft (DFG) and DFG Grants FL 153/10-1 and SFB894 (to V.F.). The histopathology and cell sorting core facilities at NYU School of Medicine are supported in part from NIH Grant UL1TR00038 from the National Center for Advancing Translational Sciences (NCATS) and by a grant (P30CA016087) to the Laura and Isaac Perlmutter Cancer Center. Analysis of murine *Orai1*, *Orai2* and *Orai3* mRNA expression was accomplished in part using data from the ImmGen Consortium. *Orai1$^{fl/fl}$* mice and 2C1.1 antibody were kindly provided by Dr H. McBride (Amgen Inc.). Embryonic stem cells used to generate *Orai2$^{LacZ/+}$* mice were generated by the trans-NIH Knock-Out Mouse Project (KOMP) and obtained from the KOMP Repository (http: //www.komp.org), which were funded by NIH Grants U01HG004085 (to Velocigene at Regeneron Inc.) and U42RR024244 (to the KOMP Repository at UC Davis and CHORI).

## Author contributions

M.V. and S.F. designed, analysed and interpreted experiments and wrote the manuscript. J.Y., I.Z., C.K., M.E., U.K., P.K.J., V.F., R.S.L., I.K. and M.Y. contributed experiments, reagents and interpreted results. M.P. analysed data, interpreted results and edited the manuscript.

## Additional information

**Competing financial interests:** S.F. is a cofounder of Calcimedica. P.K.J. is a scientist and I.K. is founder and CEO of ImmunoGenes. The other authors declare no competing financial interests.

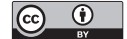

