## [Peer Review File · Nature Communications]

Reviewers' comments:

Reviewer #1 (Remarks to the Author):

The manuscript by Vaeth et al. describes the effects of genetic deletion of the store-operated calcium channel ORAI2 on the calcium influx and T cell immune response in mice lacking *Orai2*, *Orai1* or both *Orai1* and *Orai2* genes. This is the first study to use a genetically modified *Orai2* mouse model, which by replacing exon 2 with LacZ beautifully tracks expression of ORAI2 and analyzes in detail and with a high quality of data the effect of loss of ORAI2 on the immune response. However, the finding that SOCE is increased in the absence of ORAI2 is not entirely "unexpected" as stated in the abstract given several manuscripts already described an increase in SOCE upon knock-down of ORAI2, some of those references are "buried" in the discussion (Inayama M et al 2015, Alansary et al 2015 but see also Kito et al. 2015). The manuscript contains a wealth of immunological data and convincingly demonstrates how murine T cell differentiation and function indeed relies on both channel components. Especially because little is known about the physiological role of ORAI2, this data is important but the manuscript would gain additional enthusiasm from a more mechanistic insight into how ORAI2 acts as a negative regulator in some (naïve T cells) but not in effector T cells (see below). If ORAI2 prevents detrimental T cell activation then the time course of EAE development or number of Th17 cells might show an effect in the single O2 knock outs.

Suggested improvements:

1. One concern regards the fact that two gene loci exist for ORAI2 in mouse, one on chromosome 5G2 and a second on chromosome 16C1 (Wissenbach et al., 2007). Here Figure legend 1 states "The chromosomal locus 5G2 consists of five predicted exons which give rise to the splice variants ORAI2Long (ORAI2L) and ORAI2Short (ORAI2S). Alternative splicing of transcripts involves the exons 2 and 3. Since the start methionine of the 264 amino acids ORAI2L protein is localized in exon 3, the skipping of exon 3 leads to the N-terminal truncated variant ORAI2S of 250 amino acids. Its start methionine is encoded by exon 4 and corresponds to the methionine on position 15 of ORAI2L". Can the authors be sure that their targeting strategy removes all splice variants of ORAI2? Would the antibody used in Supplemental Fig.2 detect all splice variants?
2. The argument that knock-down of ORAI2 in T effector cells shows no increase in SOCE due to its lower expression is rather weak, given that the decrease shown in Fig. 1b is very small. Please show this by analyzing ORAI1 and ORAI2 protein level as done for ORAI2 in BMDMs in Suppl Figure 2. Could splicing of ORAI2 lead to altered contribution of ORAI2 upon differentiation? Fig. S2 shows some lower bands. This can be tested by splice specific primers.
3. In Figure 3 and in the model shown in Figure S8, the authors postulate a negative regulatory function due to heteromultimerization leading to a less efficient channel pore. However, it is also possible that ORAI2 homomultimers compete with ORAI1 homomultimers for limited STIM1. Does overexpression of STIM1 "rescue" the effect of ORAI2 deletion?
4. Also regarding Figure 3: Is phagocytosis/bacterial killing affected in O2^{-/-} macrophages?
5. Vice versa: Does overexpression of ORAI1 in naïve T cells (WT and O2^{-/-}) prevent the increase in SOCE seen in ORAI2^{-/-} cells? (by shifting the O1/O2 ratio towards effector cells)? This is important, as much of the discussion of the phenotype relies on this argument.
6. Is the cytokine production of Th2 cells also affected by the single/double knock-out?
7. The authors state "Taken together, our data show that both ORAI1 and ORAI2 are required for TFH

cell dependent production of antiviral antibodies and that only complete lack of SOCE compromises humoral immunity whereas residual SOCE in ORAI1- or ORAI2-deficient CD4+ TFH cells is sufficient for humoral immunity." But since ORAI2 deficient CD4 TFH cells should show an increased SOCE, it should not be a "residual" SOCE and why is the response not larger?

8. This seems to be the case for the investigation of GvHD shown in Figure 8. Here the cumulative survival is reduced in ORAI2-/- mice although none of the parameters measured in Fig 8 a-d is increased. How do the authors explain this discrepancy?

Minor points

„mRNA levels of mouse ORAI3 are higher than those of ORAI1 and ORAI2 in solid organs of mice but lower in immune cells 28“

This is not a correct statement, see skeletal muscle, B cells and other examples, when checking biops.

The real-time PCR results are all normalized to only a single house-keeping gene (HPRT). The accuracy of the results obtained by this method strongly depends on accurate transcript normalization using stably expressed genes and would benefit from the use of several HKG.

Reviewer #2 (Remarks to the Author):

The authors present a most interesting account of the possible role of the Orai2 channel in the function of T cells. The studies using a novel reporter mouse are important in revealing that Orai2 knockout has a surprising enhancing effect on calcium signals, and may have a specific role in controlling certain T cell function. The results reveal that Orai2 is highly expressed in naïve T cells whereas, in contrast, it is downregulated in effector T cells. Interestingly, deletion of Orai2 enhances both store-induced Ca²⁺ entry and CRAC currents in T cells, which is quite different to the knockout of Orai1. This is a novel and interesting finding. The results are interpreted to suggest that Orai2 acts as a negative controller of the function of CRAC channels. The study shows that Orai1 and Orai2 are both necessary to mediate store-operated Ca²⁺ entry and T cell function, both in vitro and in vivo. The deletion of both Orai1 and Orai2 interfered with responses against viral infection and prevented autoimmune inflammation. The paper draws the interesting conclusion that both Orai1 and Orai2 control store-operated Ca²⁺ entry and T cell immune responses, and that by attenuating the function of Ca²⁺ entry responses formed by Orai1 subunits, Orai2 may fine-tune the magnitude of Ca²⁺ entry signals.

The paper is extremely well written and makes some important points that would be of interest to many in immunology and signaling fields. It is worthy of consideration in a high impact journal. Nevertheless, there are some significant questions that need to be addressed by the authors.

Major Point:

1. The one major weakness of the paper is the lack of a mechanistic correlation to support the idea that Orai2 is modifying the action of Orai1 and/or that they form a complex in which the channel itself is inhibited. Thus, the cornerstone conclusion of the study is that Orai2 knockout leads to an increase in authentic CRAC current by removing some constraint from the action of Orai1. The final conclusion of the paper is to presume this occurs through heteromeric Orai1-Orai2 channel assembly that has attenuated channel function. The paper would have been stronger had some in vitro analysis been undertaken to correlate such interactions using overexpressed Orai1 and Orai2 proteins, examining with

FRET to assure that the attenuating effect of Orai2 was not through altered STIM protein interactions, and perhaps to have examined concatemers of Orai1 and Orai2 to directly observe the effects of heteromeric channels. Instead, conclusions on the inhibitory action of Orai2 is more conjecture.

Other points:

2. The study does show that in the absence of Orai2 there is no increase in message for Orai1 in the Orai2^{-/-} T cells. Critical to know is whether the increase CRAC current activity is related to any change in the presence of Orai1 channels in the plasma membrane. The authors did look at this in human fibroblasts after Orai2 knockdown (which do show increased Ca²⁺ entry), but it would have been important to examine Orai1 protein expression at the cell surface in the mouse model T cells.

3. In Fig. 3d, the data seems to be at odds with what was published previously by the Kinet lab that when Orai1 is knocked down the level of Orai2 expression increases.

4. The data in Fig. 3b on Ca²⁺ levels in the ER does show a clear increase in the Orai2^{-/-} cells which seems to be inconsistent with the statement in the text that there is no difference.

5. In Fig. 3c and d, the Mn²⁺ quench experiments are not very convincing and the summary data looks to be rather on the edge of significance.

6. The use of 100 μM La³⁺ in Fig. 3 does not tell us about the sensitivity to La³⁺ as described in the text. Also, considering the use of BTP2 and 2-APB in Fig. 3, are the authors implying that BTP2 shows a statistically significant difference in Mn²⁺ quench data in Fig. 3c between WT and Orai2^{-/-}, but not for the effect of 2-APB in patch clamp data in Fig. 3e. Also, the legends and figures only mention the concentration of La³⁺ but do not specify the concentrations used of BTP2 or 2-APB which would be important to know.

7. In Fig. 3d the currents seem to be different showing more depotentiation in DVF for Orai2^{-/-}. Is this just a random difference or is this significant? Is this related to just having Orai1 in the Orai2^{-/-}?

Overall, this is a very detailed and interesting study on the possible role of Orai2 in T cell function, but there are some weaknesses that need to be addressed in order to consolidate the significance and message of the paper. In particular, addition of mechanistic correlates could further enhance the impact of the results.

Reviewer #3 (Remarks to the Author):

Vaeth et al present their work examining T cell responses in the newly developed Orai2 knockout mouse model. This paper compares T cell functions in WT, Orai1CD4-cre, Orai2 KO and double mutant Orai1CD4-cre/Orai2 KO mice. The authors find that SOCE is actually enhanced in naive T-cells from Orai2 KO mice, which is surprising. While there is no major defect in T-cell development in any of the mice, over time, the double mutants develop splenomegaly and have lower numbers of Treg cells. In three different disease models the double mutants show profound defects in T-cell function.

I wish I could review papers like this all the time! Congratulations to Dr. Feske, his labmates and collaborators. This is simply outstanding science. Great job. Highly novel, very interesting and important work.

I have a few tiny thoughts that the team may wish to consider:

1) In the Discussion, the authors attempt to address why the Orai2 deficient T-cells don't show evidence of hyper-reactivity, even though they have increased SOCE. They argue that is because most of the disease models involve effector T-cells, which naturally down regulate Orai2, so they should be less effected in the Orai2 KO mice. They state that SOCE is actually normal in Orai2 KO effectors compared to Orai2 naive T cells. But that data isn't in the paper, I think. It would be a very easy experiment to do and should be in Fig. 2. One would expect that SOCE in naive cells to be higher than in effector cells from Orai2 KO, and that this would be reversed in WT cells. Actually, I just realized that Fig S4 partially addresses this prediction, since WT cells and Orai2 cells, stimulated with CD3/CD28 in vitro, show equal SOCE. But wouldn't it be easier and more direct to do this by FACS on fresh spleen cells, gating on naive versus effector cells?

2) If I had a fault about the overall design of this study it is with use of Orai1flx/CD4-cre mice, comparing to the Orai2 global KO. It would have been cleaner to use double flx mice or double KO mice (the later of which can't be done, since I think the Orai1 KO is a lethal mutation). But the authors constantly make conclusions for all T-cells (instead of just CD4 T cells) using the Orai1flx/CD4-cre strain. The CD8 T cells in these mice should be normal. Perhaps the authors should think about changing the wording a bit in the paper to avoid making general statements about all T cells when referring to the Orai1flx mice, and/or consider discussing this possible complication/qualification of comparing and combining a CD4 specific KO with a global KO.

3) I think the splenomegaly phenotype of the double KO mice isn't quite well described. The authors conclude that decreased number of Tregs are involved, based on Fig. 4C. But that only shows percentages, not numbers. Since the double KO has splenomegaly, the total numbers of Tregs may not be that different across all the genotypes. Additionally, the authors state that the splenomegaly is likely due to lymphoproliferation, but Fig. S5 shows roughly equal percentages of T cells subsets in spleen (and no data is presented on cell numbers). I suspect that the splenomegaly is due to accumulation of myeloid cells in the spleen. If the authors have some data on myeloid cells in the double mutant, that would be good. If not, they should at least not confuse percentages with total numbers for the Tregs. Have they investigated other aspects of autoimmunity?

Still, I think this is a great paper and a really important contribution to the field of calcium signaling (in immune cells and all cells in general). The next big step will be to test Orai2 function in human T cells doing knockdowns or other approaches. I bet the Feske lab is already on it!

REVIEWERS' COMMENTS:

Reviewer #1 (Remarks to the Author):

Thanks to the authors for the additional efforts that added significant mechanistic insight and makes this manuscript outstanding. I have no further comments.

Reviewer #2 (Remarks to the Author):

The authors have addressed my questions and criticism extremely well. In particular, they addressed the questions of the dual actions of Orai1 and Orai2, very convincingly!

Reviewer #3 (Remarks to the Author):

Great revision of a fine paper. And thanks for educating me about the CD4-cre (which I should have known).

To the editors,

We thank the editors and reviewers for carefully evaluating our manuscript "**ORAI2 modulates store-operated calcium entry (SOCE) and T cell-mediated immune responses**" (NCOMMS-16-17054) and for the constructive feedback. We appreciate all comments, which have helped to improve our manuscript. We are now resubmitting a significantly revised version of the original manuscript, in which we address the reviewers' critiques in detail. We added new mechanistic data to show that ORAI1 and ORAI2 form heteromeric channels in T cells, which was one of the major critiques of the study.

Summary of major changes to the manuscript:

1. **New Fig. 4** shows that ORAI1 and ORAI2 form heteromeric channels in primary mouse T cells and in HEK293 cells. (i) Expression of a pore-dead ORAI1 mutant in T cells from *Orai1*^{-/-} mice blocked residual SOCE mediated by ORAI2; likewise expression of a pore-dead ORAI2 mutant in T cells from *Orai2*^{-/-} mice blocked SOCE mediated by ORAI1. These results were independent of STIM1 binding and strongly suggest that ORAI1 and ORAI2 assemble in the same CRAC channel pore complex. (ii) Expression of a pore-dead ORAI1 mutant in HEK293 cells together with ORAI2 blocked CRAC currents emanating from co-expressed ORAI2. (iii) In addition, our patch clamp data show that steady state currents of ORAI1 are larger than those of ORAI2 and inactivate more slowly, providing an explanation for increased SOCE and CRAC currents in ORAI2-deficient cells compared to wildtype controls.
2. Updated **Fig. 5b,c** shows that ORAI1 and ORAI2 also regulate T_H2 cell function
3. Updated **Fig. S5 (new S5e)** provides total numbers of Treg and other immune cells in spleen and lymph nodes of *Orai1*^{fl/fl} *Cd4Cre*, *Orai2*^{-/-} and *Orai1*^{fl/fl}*Orai2*^{-/-} *Cd4Cre* (DKO) mice, explaining the splenomegaly and lymphadenopathy of the DKO mice.
4. We updated **Fig. S2a** and used the same numbering of exons in the mouse *Orai2* gene locus as reported by Wissenbach *et al.* (2007).
5. The Results and Discussion text has been edited to accommodate these changes and in response to the reviewers' critiques.
6. The title has been shortened to "*ORAI2 modulates store-operated calcium entry (SOCE) and T cell-mediated immunity*" by replacing "immune responses" with "immunity".

A detailed point-by-point response to the reviewers' critiques is provided here.

Reviewer #1

Suggested improvements:

1. One concern regards the fact that two gene loci exist for ORAI2 in mouse, one on chromosome 5G2 and a second on chromosome 16C1 (Wissenbach *et al.*, 2007). Here Figure legend 1 states "The chromosomal locus 5G2 consists of five predicted exons which give rise to the splice variants ORAI2Long (ORAI2L) and ORAI2Short (ORAI2S). Alternative splicing of transcripts involves the exons 2 and 3. Since the start methionine of the 264 amino acids ORAI2L protein is localized in exon 3, the skipping of exon 3 leads to the N-terminal truncated variant ORAI2S of 250 amino acids. Its start methionine is encoded by exon 4 and corresponds to the methionine on position 15 of ORAI2L". Can the authors be sure that their targeting strategy removes all splice variants of ORAI2? Would the antibody used in Supplemental Fig.2 detect all splice variants?

Response: We thank the reviewer for bringing up this point and have updated the cartoon in revised **Fig. S2a** depicting the exon-intron structure of the mouse *Orai2* gene and targeting strategy to match the data reported by Wissenbach *et al.* 2007. The long ORAI2 isoform (ORAI2L) reported by Flockerzi and colleagues is encoded by exons 3, 4 and 5; the short isoform (ORAI2S) is encoded by exons 4 and 5. Our targeting strategy deletes exon 4 and the 5' of exon 5 of *Orai2* and therefore abolishes expression of ORAI2S protein entirely. It also abolishes the vast majority of ORAI2L including all transmembrane domains and the C terminus and (theoretically) results only in expression of an N-terminal peptide consisting of 14 amino acids that cannot form a channel. Furthermore, we do not detect any ORAI2 protein, either ORAI2L or ORAI2S (which only differ by 14 amino acids or <2 kDa) by Western blot in macrophages (Fig. S2f) or dental enamel cells from *Orai2*^{-/-} mice (**Fig. R1** for reviewers). These results were confirmed using two different antibodies: one was described by Gross *et al.* (2007, JBC 282:19375) and Wissenbach *et al.* (2007, Cell Calcium 42:439) that recognizes a C terminal epitope of ORAI2 shared by ORAI2L and ORAI2S. It is noteworthy that the authors state in their paper that “the 14 amino acid difference of ORAI2L and ORAI2S is not resolvable” and thus no size difference is detectable using this antibody. The other antibody is a commercial antibody against ORAI2, which also recognizes the shared C terminus of ORAI2 (Abcam ab180146).

Figure R1. ORAI2 protein is undetectable in *Orai2*^{-/-} enamel organ cells. Western blot of enamel organ cells isolated from wildtype and *Orai2*^{-/-} mice. 40 µg per lane of total protein was fractionated and ORAI2 expression was detected using a rabbit monoclonal anti-ORAI2 antibody against the C-terminus of ORAI2 (Abcam ab180146). Anti-actin served as loading control. No signal for ORAI2 could be detected in *Orai2*^{-/-} enamel cells even after longer exposure.

In addition, mRNA transcripts for ORAI2L and ORAI2S that we amplified by PCR from wildtype T cells were absent in T cells from *Orai2*^{-/-} mice (**Fig. R2** for reviewers), suggesting that transcripts encoding both isoforms were abolished by our targeting strategy.

Figure R2. Ablation of exon 4 and 5 of the *Orai2* gene deletes both ORAI2L and ORAI2S isoforms. Reverse transcriptase PCR analysis of ORAI2L and ORAI2S expression in naïve T cells (*T*_{naive}) and in vitro differentiated effector T cells (*T*_{effector}) from wildtype and *Orai2*^{-/-} mice. The long isoform of ORAI2 (ORAI2L) uses an ATG as translational start codon within exon 3. Splicing may remove exon 3 and thus the ATG. A second ATG in exon 4 serves as translation start site resulting in a 14 amino acid shorter ORAI2 protein (ORAI2S). Specific primers localized in exon 2 (ORAI2S) and exon 3 (ORAI2L) with a common reverse primer in exon 4 were used to amplify both transcripts similar to Wissenbach *et al.* 2007 and Gross *et al.* 2007. See also Fig. S2a.

2. The argument that knock-down of ORAI2 in T effector cells shows no increase in SOCE due to its lower expression is rather weak, given that the decrease shown in Fig. 1b is very small. Please show this by analyzing ORAI1 and ORAI2 protein level as done for ORAI2 in BMDMs in Suppl Figure 2. Could splicing of ORAI2 lead to altered contribution of ORAI2 upon differentiation? Fig. S2 shows some lower bands. This can be tested by splice specific primers.

Response: We attempted to measure ORAI1 and ORAI2 protein levels in T cells by Western blot as suggested by the reviewer. While we can detect ORAI2 by WB in macrophages (Fig. S2f) and enamel cells (**Fig. R1** for reviewers), we did not detect an ORAI2 (or ORAI1 band for that matter) in T cells from WT mice

using two different antibodies (those described in our previous response above). A potential explanation is that ORAI2 levels are much higher in macrophages than in T cells (compare **Fig. R3** for reviewers). Therefore we cannot make a decisive statement about ORAI2 protein expression levels in naive and effector T cells.

We agree that the decrease in ORAI2 expression (or more correctly FDG levels) in Fig. 1b and ORAI2 mRNA levels in Fig. 1c from naive to effector T cells is fairly small but consistent (see also **Fig. R4** for reviewers). Simultaneous with the decrease in ORAI2 is an increase in ORAI1 levels. Therefore, the ratio of ORAI1 to ORAI2 expression changes in naive (day 0 in culture) compared to effector (day 3 in culture) T cells shown in Fig. 1e and is therefore more relevant than the reduction of ORAI2 seen in isolation.

We tested the expression of ORAI2 splice variants in naive T cells and effector T cells (**Fig. R2** for reviewers) using PCR primers that specifically detect the ORAI2L and ORAI2S splice forms. We did not detect a change in expression of either ORAI2L or ORAI2S in naive compared to effector T cells, thus making it very unlikely that "*splicing of ORAI2 leads to altered contribution of ORAI2 upon differentiation*".

3. In Figure 3 and in the model shown in Figure S8, the authors postulate a negative regulatory function due to heteromultimerization leading to a less efficient channel pore. However, it is also possible that ORAI2 homomultimers compete with ORAI1 homomultimers for limited STIM1. Does overexpression of STIM1 "rescue" the effect of ORAI2 deletion?

Response: We addressed this question with new experiments shown in **new Fig. 4**, which support the conclusion that ORAI1 and ORAI2 indeed form heteromeric channel complexes in which the presence of ORAI2 has an inhibitory effect on SOCE.

We show this by overexpression of wildtype ORAI2 (ORAI2^{WT}) in T cells from *Orai2*^{-/-} mice, which only have functional ORAI1 channels and strong SOCE; expression of ORAI2^{WT} results in suppression of SOCE (**new Fig. 4b**). In a complementary experiment, expression of wildtype ORAI1 (ORAI1^{WT}) in T cells from *Orai1*^{-/-} mice, which only have functional ORAI2 and moderate SOCE, dramatically enhances SOCE (**new Fig. 4a,d**). When we express a channel-dead mutant of ORAI1 (ORAI1^{E106Q}) in T cells from *Orai1*^{-/-} mice or a channel-dead mutant of ORAI2 (ORAI2^{E80Q}) in T cells from *Orai2*^{-/-} mice, SOCE is completely abolished (**new Fig. 4a,b**). Both experiments strongly suggest that ORAI1 and ORAI2 form heteromeric channel complexes because expression of a channel-dead mutant of one isoform (e.g. ORAI1) abolishes SOCE mediated by the other isoform (e.g. ORAI2).

To test the hypothesis mentioned by the reviewer, i.e. that "*ORAI2 homomultimers compete with ORAI1 homomultimers for limited STIM1*", we overexpressed a double-mutant of ORAI1 (ORAI1^{E106Q/L273D}) in T cells of *Orai1*^{-/-} mice (which only have functional ORAI2 channels) that is pore-dead and unable to bind to STIM1 (as shown by Muik *et al.* 2008 JBC 283:8014, Navaro-Borelly *et al.* 2008, J Physiol 586:5383 and other labs) and thus is not able to compete with endogenous ORAI2 for STIM1 binding (**new Fig. 4e-g**). Should the hypothesis of the reviewer be correct, we would have expected that expression of ORAI1^{E106Q/L273D} in T cells of *Orai1*^{-/-} mice has no effect on SOCE. This was, however, not the case because ORAI1^{E106Q/L273D} suppressed SOCE as efficiently as ORAI1^{E106Q} (**new Fig. 4d-g**). Furthermore, we also observed a suppressive effect of the STIM1 binding-deficient ORAI1^{L273D} single mutant (**new Fig. 4e,g**). This is consistent with recent data from the Lewis lab, which showed that L273D mutation in even a single subunit of a hexameric ORAI1 channel impairs CRAC channel function (Yen *et al.* 2016, Biophys J. 111:1897), arguing that each ORAI1 subunit of a homo- or heteromeric channel requires STIM1 binding. Altogether, these data strongly suggest that the inhibitory effects of ORAI2 on ORAI1 are mediated by direct interaction of both proteins in the same channel complex and not by competition for STIM1 (see **new Fig. 4h**).

This argument is further supported by additional experiments performed in HEK293 cells shown in **new Fig. 4i-n**, in which we overexpressed STIM1 together with ORAI1 or ORAI2 alone or both ORAI1/ORAI2 and examined the properties of the resulting CRAC channel currents. Co-expression of pore-mutant ORAI1^{E106A} together with ORAI2 completely abolished CRAC currents compared to expression of wildtype ORAI1, ORAI2 or both ORAI1/ORAI2. This potent dominant-negative effect occurred despite co-expression of high levels of STIM1, thus making it very unlikely that the suppressive effects of pore-dead ORAI1 are mediated by binding most or all of STIM1. The fact that ORAI1^{E106A} abolished CRAC currents mediated by ORAI2 strongly suggests that it inhibits ORAI2 through direct interaction and not by sequestering (the abundant) STIM1.

4. Also regarding Figure 3: Is phagocytosis/bacterial killing affected in O2^{-/-} macrophages?

Response: This is an interesting question because ORAI2 is highly expressed in macrophages (**Fig. R3** for reviewers). Addressing this question, we think is beyond the scope of this paper, which is focused on the mechanism of ORAI2 function and the role of ORAI1 and ORAI2 in T cell mediated immunity. It is important to note in this context that we recently published that complete abolition of SOCE in macrophages by genetic deletion of *Stim1* and *Stim2* genes in mice had no detectable effect on macrophage function including phagocytosis and bacterial killing (Vaeth *et al.* 2015, J Immunol 195:1202). From these studies we conclude that SOCE is largely dispensable for macrophage function (in contrast to its important role in T cells). It therefore was not a priority for us to investigate the impact of ORAI2 deletion on macrophage function despite the high expression of ORAI2 in these cells.

5. Vice versa: Does overexpression of ORAI1 in naïve T cells (WT and O2^{-/-}) prevent the increase in SOCE seen in ORAI2^{-/-} cells? (by shifting the O1/O2 ratio towards effector cells)? This is important, as much of the discussion of the phenotype relies on this argument.

Response: Overexpression of ORAI1 in naïve T cells is a request that is technically very difficult to fulfill because overexpression requires stimulation of T cells (for retroviral transduction) and several days in cell culture (even for transfection methods like nucleofection), and cell will not be naïve anymore. This is a real "Catch 22". We could, however, do the opposite experiment, i.e. overexpress ORAI2 in effector T cells (in which ORAI2 is normally downregulated), thereby shifting the ORAI1:ORAI2 ratio towards a ratio that is found in naïve T cells (**new Fig. 4b**). Overexpression of ORAI2^{WT} in *Orai2^{-/-}* T cells (which only express ORAI1 and have high SOCE), resulted in a strong reduction of SOCE. This result suggests that decreasing the ORAI1:ORAI2 ratio (as in naïve T cells) reduces SOCE, supporting our argument. (Of note, expression of ORAI2^{E80Q} completely abolished SOCE, indicating that the effects of ORAI2 overexpression are mediated by the formation of a heteromeric ORAI1:ORAI2 channel complex).

6. Is the cytokine production of Th2 cells also affected by the single/double knock-out?

Response: This is an interesting question and we have added data to **Fig. 6b,c** (please note updated Figure numbers) to address it. We have differentiated naïve CD4⁺ T cells from *Orai1^{fl/fl} Cd4Cre*, *Orai2^{-/-}* and *Orai1^{fl/fl}Orai2^{-/-} Cd4Cre* (DKO) mice into Th2 cells *in vitro* and restimulated them to measure expression of the 'signature' Th2 cytokine IL-4. Compared to Th2 cells from wildtype mice, Th2 cells lacking ORAI1 or both ORAI1/2 showed strongly reduced production of IL-4. No significant change in IL-4 expression was observed in ORAI2-deficient Th2 cells (**new Fig. 6b,c**). This is consistent with previous data that Th2 cells from mice with abolished or strongly reduced SOCE show impaired IL-4 production (Oh-hora *et al.* Nat immunol. 2008; McCarl *et al.* J. immunol. 2010).

7. The authors state "Taken together, our data show that both ORAI1 and ORAI2 are required for TFH cell dependent production of antiviral antibodies and that only complete lack of SOCE compromises humoral immunity whereas residual SOCE in ORAI1- or ORAI2-deficient CD4⁺ TFH cells is sufficient for humoral immunity." But since ORAI2 deficient CD4 TFH cells should show an increased SOCE, it should not be a "residual" SOCE and why is the response not larger?

Response: We thank the reviewer for pointing out this imprecise statement. SOCE is indeed not "residual", but increased in ORAI2-deficient T_{FH} cells. We have corrected this sentence (highlighted in green on page 10 of the Results section), which now reads: "... that only complete lack of SOCE compromises humoral immunity whereas partially reduced (*Orai1^{fl/fl}Cd4cre*) or increased (*Orai2^{-/-}*) SOCE in T_{FH} cells does not affect humoral immunity".

8. This seems to be the case for the investigation of GvHD shown in Figure 8. Here the cumulative survival is reduced in *ORAI2*^{-/-} mice although none of the parameters measured in Fig 8 a-d is increased. How do the authors explain this discrepancy?

Response: The reduction in "cumulative survival" of mice transplanted with allogenic *Orai2*^{-/-} T cells compared to control mice transplanted with wildtype T cells as shown in **Fig. 9f** (please note updated numbering of figures) is detectable but moderate. We agree that none of the clinical or immunological parameters we have measured (e.g. weight loss, inflammation by histology, production of IFN γ and TNF α) explains this slightly accelerated mortality of *Orai2*^{-/-} transplanted mice, which may be due to other parameters of increased inflammation we have not measured. The main conclusion of Fig. 9, however, is not that increased SOCE in *Orai2*^{-/-} T cells causes more severe GvHD, but that both ORAI1 and ORAI2 synergize to mediate SOCE, which is required for GvHD pathology.

Minor points

„mRNA levels of mouse *ORAI3* are higher than those of *ORAI1* and *ORAI2* in solid organs of mice but lower in immune cells 28“. This is not a correct statement, see skeletal muscle, B cells and other examples, when checking biogps.

Response: We thank the reviewer for pointing out this imprecise statement. We have rephrased the sentence to more accurately express what we had intended to say, which is also illustrated by **Fig. R3** for the reviewers. The sentence now reads (highlighted in green on page 3): "While expression of ORAI1 is particularly high in immune cells (including T cells) compared to non-immune cells and ORAI2 expression is high in immune cells and certain areas of the CNS, mRNA levels of ORAI3 are also abundant in many solid organs²²".

Figure R3. Expression of *Orai1*, *Orai2* and *Orai3* in different murine tissues. RNA expression data from the BioGPS (biogps.org) database of *ORAI1* (probe set 1424989), *ORAI2* (1434763), and *ORAI3* (1434064); different scales used.

The real-time PCR results are all normalized to only a single house-keeping gene (HPRT). The accuracy of the results obtained by this method strongly depends on accurate transcript normalization using stably expressed genes and would benefit from the use of several HKG.

Response: The reviewer is correct that housekeeping genes are sometimes variable and need to be selected carefully. We mainly used human and mouse *Hprt1* as it has worked well in T cells in the past (e.g. in Vaeth *et al.* 2016, *Immunity* 44, 1-15). Since some gene expression differences appear small (e.g. in Fig. 1c), we confirmed the gene expression data by repeating realtime (quantitative) PCR experiments for some genes and using additional house keeping genes besides *Hprt1* including *Actin*, *Nono* and *18S*. The results of these experiments show that normalization to these different house keeping genes yield similar results compared to normalization to *Hprt1* (see **Fig. R4** for reviewers). We conclude that *Hprt1* is a reliable housekeeping gene to test gene expression in T cells. Moreover, it is noteworthy that the ratio of *Orai1* to *Orai2* expression (as presented in Figures 1d and 1e) is not affected by the type of housekeeping gene used because *Orai1* and *Orai2* levels are normalized to the same gene. We added a sentence to the methods section that mentions these findings and reads "Additional house keeping genes (*Actin*, *Nono*, *18S*) were used to validate *Orai1* and *Orai2* expression in T cells, yielding similar results compared with *Hprt* (not shown)" highlighted in green.

Figure R4. Expression of *Orai1* and *Orai2* in naïve and effector T cells normalized to different housekeeping genes. qRT-PCR analyses of *Orai1* and *Orai2* expression in FACS-sorted CD4⁺CD62L^{hi}CD44⁻ naïve and CD4⁺CD62L⁻CD44⁺ effector T cells as shown in Fig. 1c normalized to *Hprt1* (as in Fig. 1c), *Actin*, *Nono* and *18S* as housekeeping genes. Results of *Orai1* and *Orai2* expression are comparable using these 4 housekeeping genes. Means \pm SEM of 4 mice as described in Fig. 1c.

Reviewer #2

Major Point:

1. The one major weakness of the paper is the lack of a mechanistic correlation to support the idea that *Orai2* is modifying the action of *Orai1* and/or that they form a complex in which the channel itself is inhibited. Thus, the cornerstone conclusion of the study is that *Orai2* knockout leads to an increase in authentic CRAC current by removing some constraint from the action of *Orai1*. The final conclusion of the paper is to presume this occurs through heteromeric *Orai1*-*Orai2* channel assembly that has attenuated channel function. The paper would have been stronger had some *in vitro* analysis been undertaken to correlate such interactions using overexpressed *Orai1* and *Orai2* proteins, examining with FRET to assure that the attenuating effect of *Orai2* was not through altered STIM protein interactions, and perhaps to have examined concatemers of *Orai1* and *Orai2* to directly observe the effects of heteromeric channels. Instead, conclusions on the inhibitory action of

Orai2 is more conjecture.

Response: We thank the reviewer for pointing out this limitation of the study and agree that we had not presented a mechanistic explanation how "ORAI2 is modifying the action of ORAI1". We have conducted a series of new experiments (presented in **new Fig. 4**), which demonstrate that ORAI1 and ORAI2 form a heteromultimeric channel in T cells. These new findings have been described in more detail in response to reviewer 1, critique #3 above. We briefly summarize the main points here:

- Our conclusion that "*Orai2 is modifying the action of Orai1 and/or that they form a complex in which the channel itself is inhibited*" is supported by the expression of pore-dead mutant ORAI2 (ORAI2^{E80Q}) in T cells that only express ORAI1 to mediate SOCE (from *Orai2*^{-/-} mice) or alternatively by expression of pore-dead mutant ORAI1 (ORAI1^{E106Q}) in T cells that only express ORAI2 to mediate SOCE (from *Orai1*^{-/-} mice) (**Fig. 4a,b**).
- This conclusion is furthermore supported by the fact that overexpression of a pore-dead ORAI1 mutant (ORAI1^{E106A}) and ORAI2 together with STIM1 in HEK293 cells completely abolished CRAC currents (**Fig. 4i-n**). The most likely explanation for this and the abovementioned observation is the incorporation of pore-dead mutant ORAI proteins into a heteromeric channel complex, in which they interfere with Ca²⁺ conductance in the channel pore.
- We also addressed an alternative explanation pointed out by the reviewer ("*that the attenuating effect of Orai2 was [...] through altered STIM protein interactions*"). We overexpressed a double-mutant of ORAI1 (ORAI1^{E106Q/L273D}) that cannot conduct Ca²⁺ and is unable to bind to STIM1 and thus unable to compete with ORAI2 for STIM1. We found that its expression in T cells from *Orai1*^{-/-} mice (which only have ORAI2 to mediate SOCE) almost completely suppressed SOCE similar to expression of ORAI1^{E106Q}. This inhibition should not have occurred if it was mediated by sequestering STIM1. We therefore think it is unlikely that ORAI2 inhibits ORAI1 function by sequestering STIM1 and preventing it from efficiently activating ORAI1. Please see also our response to reviewer #1 point 3.

Our electrophysiological measurements of CRAC currents in HEK293 cells expressing ORAI1 or ORAI2 (together with overexpressed STIM1) show that peak currents in ORAI1, ORAI2 and ORAI1/ORAI2 expressing cells are comparable (**Fig. 4i-k**). It seems unlikely therefore that different peak conductances can explain the differences in SOCE in *Orai1*^{-/-} and *Orai2*^{-/-} T cells. By contrast, steady state currents measured at the end of a 100 ms pulse to -100 mV were significantly larger in cells expressing ORAI1 compared to ORAI1/ORAI2 expressing cells (**Fig. 4l**). This is very likely explained by the increased fast Ca²⁺ dependent inactivation (CDI) observed in ORAI2 and ORAI1/ORAI2 expressing cells compared to those expressing ORAI1 alone (**Fig. 4m-n**). Previous studies had indeed reported more pronounced fast CDI of ORAI2 compared to ORAI1 (Lis *et al.* 2007, *Curr Biol* 17:794 and Lee *et al.* 2009, *PNAS* 106:14687), and our results indicate that heteromeric ORAI1/ORAI2 channels share the inactivation features of ORAI2 channels. A more detailed biophysical characterization of homo- and heteromeric ORAI1 and ORAI2 channel properties awaits future studies, which could include concatemers as suggested by the reviewer, but this is beyond the scope of this paper.

Other points:

2. *The study does show that in the absence of Orai2 there is no increase in message for Orai1 in the Ora2-/- T cells. Critical to know is whether the increase CRAC current activity is related to any change in the presence of Orai1 channels in the plasma membrane. The authors did look at this in human fibroblasts after Orai2 knockdown (which do show increased Ca²⁺ entry), but it would have been important to examine Orai1 protein expression at the cell surface in the mouse model T cells.*

Response: We agree that increased ORAI1 protein expression in the plasma membrane might explain increased SOCE and CRAC currents in ORAI2-deficient cells. This is why we had studied ORAI1 protein expression in human fibroblasts and found that shRNA mediated suppression of ORAI2 does not increase ORAI1 expression in the plasma membrane. The reason we did these experiments in human cells, and not in cells from *Orai2*^{-/-} mice (either T cells or macrophages), was that all available anti-ORAI1 antibodies that detect the extracellular III-IV loop of ORAI1 (and can therefore specifically detect ORAI1 at the surface of cells) are

specific for human ORAI1 and do not recognize mouse ORAI1. Since ORAI2 deletion in human fibroblasts leads to an increase of SOCE as does ORAI2 deletion in mouse T cells or macrophages, we think that given the technical limitations of detecting mouse ORAI1 at the surface of T cells testing ORAI1 expression in human fibroblasts is a valid and informative approach.

3. In Fig. 3d, the data seems to be at odds with what was published previously by the Kinet lab that when *Orai1* is knocked down the level of *Orai2* expression increases.

Response: Fig. 2d [sic] and Fig. S3c [sic] does not show a significant increase in *ORAI2* mRNA levels in shORAI1 treated human fibroblast cells (**Fig. 2d**) or ORAI1-deficient primary T cells (**Fig. S3d**). The paper from the Kinet lab the reviewer is likely referring to is Vig *et al.* (Nat Immunol. 2008, 9:89), in which the authors study mast cells and T cells of *Orai1*^{-/-} mice. In this paper (see Fig. S5 in Vig *et al.*), the authors show that ORAI2 levels are decreased [sic] in T cells of *Orai1*^{-/-} mice compared to wildtype controls. However, the experiment was repeated only twice and no error bars or statistical significance are provided, and it thus remains difficult to fully evaluate this result.

4. The data in Fig. 3b on Ca²⁺ levels in the ER does shows a clear increase in the *Orai2*^{-/-} cells which seems to be inconsistent with the statement in the text that there is no difference.

Response: We corrected the statement in the text, which now reads (highlighted in green on page 6) "Depletion of Ca²⁺ stores with ionomycin in the absence of extracellular Ca²⁺, however, showed only a very moderate increase in Ca²⁺ store content in *Orai2*^{-/-} compared to WT BMDMs (Fig. 3a,b)".

5. In Fig. 3c and d, the Mn²⁺ quench experiments are not very convincing and the summary data looks to be rather on the edge of significance.

Response: The Mn²⁺ quench experiments shown in **Fig. 3c** were repeated 3 times using cells from 3 WT and 3 *Orai2*^{-/-} mice measured at least in duplicates. While we agree that the differences appear small, they are reproducible and statistically significant (P values: WT vs *Orai2*^{-/-} = 0.0039). The increase in the Mn²⁺ quench rate, which is a measure of the rate of Mn²⁺ influx and CRAC channel function, in *Orai2*^{-/-} cells is consistent with the increase in CRAC current amplitudes in *Orai2*^{-/-} cells shown in **Fig. 3e**. We therefore consider the Mn²⁺ quench data to be correct.

6. The use of 100 μ M La³⁺ in Fig. 3 does not tell us about the sensitivity to La³⁺ as described in the text. Also, considering the use of BTP2 and 2-APB in Fig. 3, are the authors implying that BTP2 shows a statistically significant difference in Mn²⁺ quench data in Fig. 3c between WT and *Orai2*^{-/-}, but not for the effect of 2-APB in patch clamp data in Fig. 3e. Also, the legends and figures only mention the concentration of La³⁺ but do not specify the concentrations used of BTP2 or 2-APB which would be important to know.

Response: (i) We used La³⁺ purely for leak subtraction, and not to assess sensitivity to the blocker. Therefore, we have removed the reference to sensitivity of La³⁺ blockade, which appears to be identical between the different conditions at the dose used. (ii) "Are the authors implying ..." We used BTP2 for the Mn²⁺ quench experiments because we had observed a high background quench signal in non-stimulated cells and BTP2 was used to inhibit the CRAC-specific component of this background. The cause of the higher Mn²⁺ quench rate in BTP2-treated *Orai2*^{-/-} vs WT cells is not clear, but suggest that BTP2 is not an entirely specific blocker of CRAC channels as has been reported (Takezaw *et al.* 2006, Mol Pharm 69:1413). To avoid any confusion we decided to remove the data using BTP-2 in **Fig. 3c**. (iii) We apologize for the oversight and added 2-APB concentrations to the Figure legend. 2-APB in **Fig. 3e** was used at 50 μ M.

7. In Fig. 3d the currents seem to be different showing more depotentiation in DVF for *Orai2*^{-/-}. Is this just a random difference or is this significant? Is this related to just having *Orai1* in the *Orai2*^{-/-}?

Response: The increased depotentiation of CRAC currents observed using DVF in cells of *Orai2*^{-/-} mice

compared to cells in Fig. 3d seems to be random. We observed a high variability even in WT cells. Analyzing all cells tested did not yield statistically significant differences in depotentialiation between WT and *Orai2*^{-/-} cells.

Reviewer #3

I wish I could review papers like this all the time! Congratulations to Dr. Feske, his labmates and collaborators. This is simply outstanding science. Great job. Highly novel, very interesting and important work.

I have a few tiny thoughts that the team may wish to consider:

*1) In the Discussion, the authors attempt to address why the *Orai2* deficient T-cells don't show evidence of hyper-reactivity, even though they have increased SOCE. They argue that is because most of the disease models involve effector T-cells, which naturally down regulate *Orai2*, so they should be less effected in the *Orai2* KO mice. They state that SOCE is actually normal in *Orai2* KO effectors compared to *Orai2* naive T cells. But that data isn't in the paper, I think. It would be a very easy experiment to do and should be in Fig. 2. One would expect that SOCE in naive cells to be higher than in effector cells from *Orai2* KO, and that this would be reversed in WT cells. Actually, I just realized that Fig S4 partially addresses this prediction, since WT cells and *Orai2* cells, stimulated with CD3/CD28 *in vitro*, show equal SOCE. But wouldn't it be easier and more direct to do this by FACs on fresh spleen cells, gating on naive versus effector cells?*

Response: The requested data are indeed in the paper. We show the difference in SOCE between freshly isolated, mostly naive T cells (**Fig. 2a,b** and **Fig. S3**) and *in vitro*-induced effector cells (**Fig. S4**) from WT and *Orai2*^{-/-} mice. These data are consistent with a decrease in ORAI2 mRNA levels in *in vitro*-induced effector T cells compared to naive T cells and suggest that ORAI2 plays a role in SOCE in naive, but a lesser or no role in effector T cells.

*2) If I had a fault about the overall design of this study it is with use of *Orai1*flx/CD4-cre mice, comparing to the *Orai2* global KO. It would have been cleaner to use double flx mice or double KO mice (the later of which can't be done, since I think the *Orai1* KO is a lethal mutation). But the authors constantly make conclusions for all T-cells (instead of just CD4 T cells) using the *Orai1*flx/CD4-cre strain. The CD8 T cells in these mice should be normal. Perhaps the authors should think about changing the wording a bit in the paper to avoid making general statements about all T cells when referring to the *Orai1*flx mice, and/or consider discussing this possible complication/qualification of comparing and combining a CD4 specific KO with a global KO.*

Response: The reviewer is correct that conventional (i.e. complete) knockout of *Orai1* is perinatally lethal; we therefore used *Orai1*^{fl/fl} *Cd4Cre* mice to delete *Orai1* only in T cells. It is important to note, however, that *Orai1* is deleted in both CD4 and CD8 T cells of *Orai1*^{fl/fl} *Cd4Cre* mice as is apparent by reduced SOCE in CD4 and CD8 T cells in **Fig. S3a,b**. The reason for deletion in both T cell lineages is that immature T cells in the thymus pass through a stage during which they express both CD4 and CD8 ("double positive", or DP, thymocytes) before committing to become either CD4 or CD8 T cells. Therefore *Cd4Cre* can be used to delete loxP-flanked ('floxed') genes in all T cells. To compare the cell intrinsic function of ORAI1 and/or ORAI2 in T cells we made use of 2 different adoptive T cell transfer models, i.e. the transfer IBD colitis model (Fig. 8) and the allogenic GvHD model (Fig. 9). In both models, host mice receive T cells that are deficient for ORAI1 and/or ORAI2 whereas the host environment is fully wildtype.

3) I think the splenomegaly phenotype of the double KO mice isn't quite well described. The authors conclude that decreased number of Tregs are involved, based on Fig. 4C. But that only shows percentages, not numbers. Since the double KO has splenomegaly, the total numbers of Tregs may not be that different across all the genotypes. Additionally, the authors state that the splenomegaly is likely due to lymphoproliferation, but Fig. S5 shows roughly equal percentages of T cells subsets in spleen (and no data is presented on cell numbers). I suspect that the splenomegaly is due to accumulation of myeloid cells in the spleen. If the authors have some data on myeloid cells in the double mutant, that would be good. If not, they should at least not

confuse percentages with total numbers for the Tregs. Have they investigated other aspects of autoimmunity?

Response: This is an important point. We have added total cell numbers in the spleen of WT, *Orai1^{fl/fl} Cd4Cre* mice, *Orai2^{-/-}* and *Orai1^{fl/fl}Orai2^{-/-} Cd4Cre* (DKO) mice to **Fig. S5 (new Fig. S5e)**. The total numbers of B cells, CD4⁺ T cells, CD8⁺ T cells and myeloid cells are significantly increased in the spleen and lymph nodes (LNs) of *Orai1^{fl/fl}Orai2^{-/-} Cd4Cre* mice compared to the other 3 genotypes, explaining the splenomegaly of these mice. The total numbers of CD25⁺ Foxp3⁺ Treg cells are reduced in the spleen of *Orai1^{fl/fl}Orai2^{-/-} Cd4Cre* mice and unchanged in their LNs. But since the numbers of effector cells (B cells, CD4⁺ T cells and CD8⁺ T cells) are increased in LNs, the ratio of Treg-to-effector cells is decreased in *Orai1^{fl/fl}Orai2^{-/-} Cd4Cre* mice. It has been shown in several studies that this ratio is critical for the ability of Treg cells to prevent lymphoproliferation and autoimmunity (for instance: Hall BM et al. 2008, *Transpl Immunol* 18: 291; Nomura M et al. 2006, *Transpl Immunol* 15:311; Thornton AM et al. 1998, *J Exp Med* 188: 287; Thornton AM and Shevach EM 2000, *J Immunol* 164: 183).